# Calphad Modeling of LRO and SRO Using *ab initio* Data

**Masanori Enoki [1,\*], Bo Sundman [2], Marcel H. F. Sluiter [3]**  **, Malin Selleby [4] and Hiroshi Ohtani [1]**

[1]   Institute of Multidisciplinary Research for Advanced Material (IMRAM), Tohoku University, Sendai 980-8577, Japan; h.ohtani@tagen.tohoku.ac.jp

[2]   OpenCalphad, 9 Allée de l'Acerma, 91190 Gif sur Yvette, France; bo.sundman@gmail.com

[3]   Materials Science and Engineering, Delft University of Technology, 2600 AA Delft, The Netherlands; m.h.f.sluiter@tudelft.nl

[4]   Materials Science and Engineering, KTH Royal Institute of Technology, 10044 Stockholm, Sweden; malin@kth.se

\*   Correspondence: enoki@tohoku.ac.jp

**Abstract:** Results from DFT calculations are in many cases equivalent to experimental data. They describe a set of properties of a phase at a well-defined composition and temperature, $T$, most often at 0 K. In order to be practically useful in materials design, such data must be fitted to a thermodynamic model for the phase to allow interpolations and extrapolations. The intention of this paper is to give a summary of the state of the art by using the Calphad technique to model thermodynamic properties and calculate phase diagrams, including some models that should be avoided. Calphad models can decribe long range ordering (LRO) using sublattices and there are model parameters that can approximate short range ordering (SRO) within the experimental uncertainty. In addition to the DFT data, there is a need for experimental data, in particular, for the phase diagram, to determine the model parameters. Very small differences in Gibbs energy of the phases, far smaller than the uncertainties in the DFT calculations, determine the set of stable phases at varying composition and $T$. Thus, adjustment of the DFT results is often needed in order to obtain the correct set of stable phases.

**Keywords:** calphad; DFT; thermodynamic models; phase diagrams

## 1. Introduction

In materials physics the scientists performing density functional theory (DFT) calculations have a preference for the cluster variation method (CVM) or Monte Carlo (MC) methods to calculate the equilibria in their systems because these can describe both long range ordering (LRO) and short range ordering (SRO) in the crystalline phases. However, both CVM, developed by Kikuchi [1], and MC, are computationally heavy for multi-component systems, and this limits their use for real alloys. Thus there is interest in using Calphad type models for equilibrium and phase diagram calculations and this paper gives an introduction to how these models handle LRO and SRO.

The Calphad models based on the compound energy formalism (CEF) [2], as described in this paper, are computationally highly efficient and make it possible to calculate multi-component equilibria and phase diagrams for systems with many components and phases. Such phases can feature four or more sublattices to describe complex states of LRO and SRO. CEF also includes an approximate way to handle SRO [3]. The use of data from DFT calculations in Calphad type models has been discussed in many papers [4–8] and an example of how to use DFT data in CEF models will be given below together with some advice for selecting the appropriate models for different phases. Recently, a further modification of CEF has been proposed by Dupin et al. [9] to improve both extrapolations and calculation speed.

In the first section there is an introduction to the Calphad descriptions starting from pure elements. Then follows a detailed description of the reciprocal model used in CEF to handle LRO and SRO. In the third section an example of DFT calculations for the Fe–Ni system is combined with experimental data and fitted to both CEF and CVM models which can handle phases with order–disorder transitions. Section 4 gives an overview of CEF models for phases with LRO, including ionic constituents and defects with and without SRO, and in the last section is some advice on using CEF models in assessments.

*1.1. The Pure Element Data*

In order to use results from DFT calculations in a Calphad model the basic data structuring must be understood. In Calphad the pure elements in their stable state at 298.15 K and 1 bar is referred to as SER, the standard element reference, state. The Gibbs energy for an element A at $T$ above 298.15 K relative to SER is expressed as a polynomial in $T$ and $P$.

$$G_A^{SER} - H_A^{SER}(T = 298.15) = GHSERA \quad = \quad f(T, P) \tag{1}$$

up to some high $T$ limit. In the Scientific Group Thermodata Europe (SGTE) unary database [10] the low $T$ limit for $f(T, P)$ is 298.15 K and it can thus not be extended to 0 K. For elements with magnetic transitions the function GHSERA describes a para-magnetic state and the contribution to the Gibbs energy from magnetism is described using a separate model with parameters for the Curie $T$ and the Bohr magneton number, $\beta$ because in an alloy they depend on the composition. The details for this can be found in [10] and in many reference books—for example, [11].

The GHSERA functions, as shown in Equation (1), are very important also for describing the heat capacities of many intermetallic compounds in an assessment of the thermodynamic data for a system. If there are no experimental data, the Kopp–Neumann rule [12,13] can be used to estimate the heat capacity of the compound as the mean of the heat capacities of the constituent elements in the compound. In such cases the enthalpy and entropy of formation of the compound are the only properties that need to be assessed or calculated by DFT.

The fact that the unary database contains heat capacities for the pure elements makes it possible to calculate enthalpy differences at various values of $T$ as well as heat balances in multi-component systems. That is necessary for many kinds of applications in computational thermodynamics (CT).

However, the most important part of the unary database contains the estimations of the "lattice stabilities" introduced by Kaufman [14,15] for all phases in which an element may dissolve but is metastable as pure. To calculate the equilibrium in a binary, ternary or multi-component system where each phase is described by a multi-dimensional Gibbs energy hyper-surface in composition, the lattice stability values represent the end points of such a surface at the pure element for each phase. The lattice stability values are estimations which were challenged already in 1985 by Skriver [16], but in a recent review by van de Walle [17] the current values were shown to be reasonable, in particular, because DFT calculations showed that many elements are mechanically unstable at 0 K in many structures.

*1.2. The New Unary Database*

A new unary database is under development with the aim of including thermodynamic descriptions of pure elements down to 0 K, including the 3rd law for defect free crystalline phases, and improving the approximations made in 1991 for the extrapolations of the solid phases to high $T$ and the liquid phase below its solidification $T$. Some results from this work are described in [7,18–22]. In most cases this new unary database will be compatible with the current one in the stable range of the phases and does not require any significant changes in the modeling of the solution phases [20].

*1.3. Terminology*

The authors of this paper had several discussions about the terminology and finally decided to include this list to define their use in this paper (which may not be the most general).

**CEF** stands for compound energy formalism, a family of models using sublattices to describe LRO assuming ideal mixing of the constituents on each sublattice. There can be energetic interactions between the constituents.

**Cluster** is an assembly of sites in the crystal (in the context of CVM). Each possible set of constituents occupying these sites represent a cluster probability. In some models species are used to represent clusters.

**Components** are normally the elements but for convenience, to set conditions for an equilibrium calculation, they can be defined as any orthogonal subset of the species—for example, "$Al_2O_3$, CaO, O" rather than "Al, Ca, O".

**Composition** of a phase is the fraction of its components which can be calculated from its model and the fraction of its constituents or cluster probabilities. It can be per moles, mass or volume.

**Constituent** is a species in a specific phase and sublattice.

**Constituent fraction**, denoted $y_{si}$ for constituent $i$ in sublattice $s$ of a phase. When a specific value is used for either or both indices, a comma "," is used in between.

**CVM** stands for cluster variation method where the configurational entropy is evaluated using a sequence of clusters representing points, pairs, triangles, etc., in the lattice. Both LRO and SRO can be described by the clusters.

A **disordered fraction set** can be used to include parameters which do not depend on the constitution of a phase with sublattices; see, for example, Sundman et al. [23]. The disordered fractions can be calculated from the constituent fractions using Equation (2). Phases with order–disorder transitions can have separate interstitial sublattices.

**Element** is an atom in the periodic chart.

**Endmember** is used in CEF to denote a compound with a specific constituent in each sublattice of a phase.

An **ideal solution** is a model where all constituents of a phase mix randomly without any energetic interactions.

**LRO:** long range ordering in a crystalline phase depends on the arrangement of the constituents in the sublattices in the whole phase.

**Mole fraction** in a CEF model for a component A is denoted $x_A$ and calculated as:

$$x_A \quad = \quad \frac{\sum_s a_s \sum_i b_{iA} y_{si}}{\sum_B \sum_s a_s \sum_i b_{iB} y_{si}} \tag{2}$$

where $a_s$ is the site ratio, $y_{si}$ is a constituent fraction and $b_{iA}$ is a stoichiometric factor.

**Phase** has a defined lattice or is a gas, liquid or amorphous phase and has a thermodynamic model.

**Reciprocal model** within CEF has two sublattices with two constituents in each.

**Site ratio**, denoted $a_s$ for sublattice $s$, is the number of sites in the sublattice in the unit cell.

**Species** is a molecule-like aggregate with one or more elements with fixed ratios—for example, $H_2O$. The **vacancy**, denoted Va, is a special species used to represent a vacant site in a phase. A species may also have an electric charge and be called ion.

**SRO:** Short range ordering is present when the local environment of a given species is not randomly surrounded by other species. However, the deviation from randomness vanishes as the distance increases.

**Stoichiometric factor**, denoted $b_{iA}$ is the ratio of element A in species $i$.

**Sublattice:** A specific set of sites in the unit cell of the lattice. Only sites which are, at least partially, occupied by atoms in a crystal structures are recognized as sublattices. In some cases a sublattice may combine sites which are crystallographically different.

A **substitutional regular solution** is a model where all constituents are assumed to mix randomly with energetic interaction parameters between sets of two or more constituents.

The **thermodynamic model** describes the Gibbs energy of a phase as a function of $T, P$ and its constitution or cluster probabilities.

*1.4. Modeling Long and Short Range Ordering*

In the CVM, configurational entropy associated with SRO and LRO is treated approximately within a certain spatial range defined by the so-called maximal clusters. A well-known set of maximal clusters with a spatial range including both nearest and next-nearest neighbors are the octahedron-tetrahedron clusters for the fcc structure and its superstructures. The maximal clusters contain smaller subclusters. The configurational entropy is then evaluated as a sum of entropy contributions over all clusters from the maximal clusters down to the smallest subclusters. The complicating factor that occurs in this sum, is that smaller subcluster contributions are already included to various extents in the larger clusters. Therefore, a sophisticated weighting scheme through the so-called Kikuchi–Barker coefficients is required to assure that each entropy contributions enters in the sum with a proper weight. The selection of the maximal clusters is not trivial. There exist maximal clusters that do not yield properly weighted entropy sums; however, well-chosen maximal clusters can give remarkably accurate configurational entropy values. The CVM is not well suited for treating cases with many species, or when long-ranged interactions necessitate large maximal clusters. In both these cases the number of (sub)cluster configurations becomes overwhelmingly large and the numerical solution of the CVM equations impractical. For that reason the CEF is the method of choice within the Calphad approach.

In the CEF, a phase with several sublattices (sites in the unit cell) is thought of as a mixture of several endmembers that at least hypothetically can exist and which specifies a constituent on each sublattice. The configurational entropy of the phase is calculated by assuming that the constituents on each sublattice mix randomly weighted by the site ratio for each sublattice. This can be considered as the LRO contribution to the configurational entropy of the phase.

For a phase which always has LRO and would require complex clusters in the CVM, it is possible to use a CEF model and adjust some of the endmember energies to compensate for the lack of SRO. In some cases it may be necessary to choose a cell that is a multiple of the unit cell, requiring more endmember energies to be adjusted or to use a method discussed by Liu [24]. For very small unit cells, for example, in phases with order–disorder transitions such as A1/L1$_2$ or A2/B2, which can exist with considerable SRO contributions without any LRO (and thus without additional endmember parameters in a CEF model), an approximate SRO contribution can be added, as explained in Section 2.4. Clearly, the CEF model provides an easily solvable model that requires as input a few fraction variables pertaining to sublattices, and a limited number of endmember energies. The endmember energies can be computed with DFT methods, and in Section 5 the methodology to perform an assessment combining experimental and DFT data is discussed.

## 2. Using the Reciprocal Model in CEF to Model LRO and SRO

CEF and CVM are mean field models which both aim to describe the thermodynamic properties of crystalline phases. Nevertheless they are based on completely different concepts: CEF originates from a model describing the entropy of molten salts without any crystalline structure. The reciprocal model, which is the basic modeling tool within CEF, is based on studies of reactions in salt systems such as:

$$NaCl + KBr \quad <=> \quad NaBr + KCl$$

and Temkin [25] proposed in 1945 a model mixing cations and anions on two separate sets of sites, a two-sublattice model without any SRO. Hillert and Staffanson [26] adapted this model 1970 to describe interstitial solutions of C in steels and it was generalized in 1981 to multiple sublattices and

constituents by Sundman and Ågren [27] as part of a long-term project to develop databases for steels, oxides and superalloys. In 1998 an approximate SRO contribution was added to CEF by comparing the configurational entropies for a four-sublattice CEF model and a tetrahedron CVM model for FCC in an assessment of Au–Cu by Sundman et al. [3]. In 2001, Hillert [2] summarized this development which has also resulted in the Thermo-Calc [28] software and databases.

All CEF models with sublattices are extensions of the reciprocal system with two sublattices and two constituents in each, denoted:

$$(A, B)_a (C, D)_c$$

where A and B are constituents on a sublattice with the site ratio $a$ and C and D are constituents on the other sublattice with the site ratio $c$. Note that this model does not assume any geometric arrangement of the sites. The same species can be constituents in both sublattices and the site ratios are the only references to the crystal structure.

The reciprocal model is important to understand because phases with several sublattices and with three or more constituents in each will have many model parameters, particularly endmembers. To understand the relations between the endmembers it is useful to divide the phase into a sequence of reciprocal subsystems. See, for example, how to find the endmembers related to energy for a Frenkel defect in Section 4.3. We thus start by exploring the properties of the reciprocal model, and for further details of the CEF model, see the book by Hillert [29] or Lukas et al. [11].

The constitutional square of the reciprocal model is shown in Figure 1a with the four endmembers (A:C), (A:D), (B:C) and (B:D) in the corners and four interaction parameters, $L_{AB:C}$, etc., along the edges. In the center there is a reciprocal interaction parameter, $L_{A,B:C,D}$, representing interactions on both sublattices. In Figure 1b an example of the Gibbs energy surface for a reciprocal system is shown.

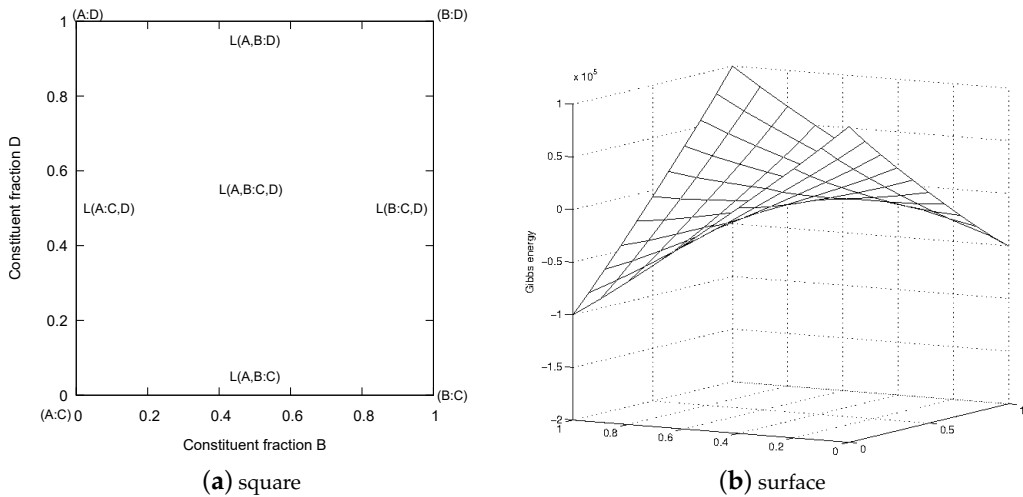

**Figure 1.** The constitutional square of a reciprocal system in (**a**) and an example of its Gibbs energy surface in (**b**).

The Gibbs energy for a reciprocal model for the phase $\alpha$ is:

$$G_M^\alpha \;=\; {}^{\text{srf}}G_M^\alpha - T\,{}^{\text{cfg}}S_M^\alpha + {}^{E}G_M^\alpha \tag{3}$$

where $G_M$ is the Gibbs energy per mole formula unit and $T$ the absolute temperature. The terms ${}^{\text{srf}}G_M$ (the surface of reference), ${}^{\text{cfg}}S_M$ (configurational entropy) and ${}^{E}G_M$ (excess Gibbs energy) are given in Equations (4)–(6). The subscript $M$ in Equation (3) is used to indicate moles per formula units of the

phase because a constituent may contain more than one atom or represent a vacancy, whereas $m$ in $G_m$ means per mole of components. The phase superscript is omitted in the following unless needed.

$$
\begin{aligned}
{}^{\mathrm{srf}}G_M &= y_{1,\mathrm{A}}y_{2,\mathrm{C}}\,{}^{\circ}G_{\mathrm{A:C}} + y_{1,\mathrm{A}}y_{2,\mathrm{D}}\,{}^{\circ}G_{\mathrm{A:D}} + y_{1,\mathrm{B}}y_{2,\mathrm{C}}\,{}^{\circ}G_{\mathrm{B:C}} + y_{1,\mathrm{B}}y_{2,\mathrm{D}}\,{}^{\circ}G_{\mathrm{B:D}} & (4)
\end{aligned}
$$

$$
{}^{\mathrm{cfg}}S_M = -R(a(y_{1,\mathrm{A}}\ln(y_{1,\mathrm{A}}) + y_{1,\mathrm{B}}\ln(y_{1,\mathrm{B}})) + c(y_{2,\mathrm{C}}\ln(y_{2,\mathrm{C}}) + y_{2,\mathrm{D}}\ln(y_{2,\mathrm{D}}))) \tag{5}
$$

$$
\begin{aligned}
{}^{E}G_M &= y_{1,\mathrm{A}}y_{1,\mathrm{B}}(y_{2,\mathrm{C}}L_{\mathrm{A,B:C}} + y_{2,\mathrm{D}}L_{\mathrm{A,B:D}}) + y_{2,\mathrm{C}}y_{2,\mathrm{D}}(y_{1,\mathrm{A}}L_{\mathrm{A:C,D}} + y_{1,\mathrm{B}}L_{\mathrm{B:C,D}}) + \\
&\quad y_{1,\mathrm{A}}y_{1,\mathrm{B}}y_{2,\mathrm{C}}y_{2,\mathrm{D}}L_{\mathrm{A,B:C,D}} \tag{6}
\end{aligned}
$$

where $y_{si}$ in the fraction of constituent $i$ on sublattice $s$; ${}^{\circ}G_{\mathrm{A:C}}$ is the Gibbs energy of the endmember (A:C), where the "pre-superscript" ${}^{\circ}$ means the parameter is independent of the constitution. $R$ is the gas constant and ${}^{\mathrm{cfg}}S_M$ the configurational entropy assuming ideal mixing on each sublattice. The contribution depends on the number of sites $a$ and $c$ on each sublattice. The interaction parameters $L_{\mathrm{A,B:C}}$, etc., give the interaction between two constituents in one sublattice with a specific constituent in the second sublattice. Finally $L_{\mathrm{A,B:C,D}}$ is an interaction parameter for all four constituents, and all interaction parameters may depend on $T$.

The interaction parameters $L_{\mathrm{A,B:C}}$ may depend on the fractions of the interacting constituents as a Redlich–Kister series:

$$
L_{\mathrm{A,B:C}} = \sum_{\nu=0}^{n}(y_{1,\mathrm{A}} - y_{1,\mathrm{B}})^{\nu} \cdot {}^{\nu}L_{\mathrm{A,B:C}} \tag{7}
$$

where $\nu$ is a power and $n$ should normally not exceed three and each ${}^{\nu}L_{\mathrm{A,B:C}}$ may depend linearly on $T$. The reciprocal parameter, $L_{\mathrm{A,B:C,D}}$, represents a simultaneous exchange energy between all four constituents and is frequently used to approximate the SRO contribution, as discussed in Section 2.4.

Understanding the properties of the reciprocal system is essential for modeling any multi-sublattice phase and it occurs in many different phases; for example:

- An interstitial solution of C in the FCC phase in a C–Fe–Ti system is modeled as $(\mathrm{Fe,Ti})_1(\mathrm{C,Va})_1$; see Section 4.1.
- A Laves phase with anti-site defects, $(\mathrm{A,B})_2(\mathrm{A,B})_1$, and other topologically close-packed (TCP) phases; see Section 4.2.1.
- BCC ordering as in Fe–Al or Al–Ni; see Sections 2.3 and 4.5.3.
- FCC ordering in Fe–Ni and Al-Ni; see Sections 2.3, 2.6 and 4.5.2.
- Oxides and other metal-nonmetal phases modeled with two or more sublattices; see Section 4.3.
- Phases with defects; see Section 4.4.

## 2.1. The Endmember Concept and Metastability

An endmember parameter ${}^{\circ}G_{\mathrm{A:C}}$ represents the Gibbs energy of a compound with a given structure. These are sometimes stable compounds, but in a solution phase they most frequently are metastable at all $T$. For example, in a Laves_C14 phase modeled as

$$
(\mathrm{Fe,Mo})_2(\mathrm{Fe,Mo})_1
$$

only the endmember, $\mathrm{Fe_2Mo}$, represents a stable phase. Two of the other endmembers represent pure Fe and pure Mo as metastable in the Laves_C14 structure and the fourth endmember is a metastable antistructure Laves_C14 phase with the wrong stoichiometry.

The Laves_C14 phase is an example of a phase with very small solubility in the Fe–Mo system and it is frequently modeled using defects. The defects can be of different types, such as vacancy, anti-site atom or interstitial. In Calphad models the Laves phases are normally modeled with anti-site atoms as Laves phases occur in many other binary systems, including either Fe or Mo. In order to

develop multi-component databases it is necessary to treat all phases with the same structure with the same model because the binary Laves phase often dissolves other elements. It may even occur inside a ternary system without being stable in any of the binaries. It is thus simpler to use the same type of defect in all cases, especially to reduce the number of endmembers. In order to model the Laves phase in a particular binary with no intention to extend to multi-component system, one should of course use the physically correct defects.

The enthalpy of formation of the metastable endmembers for many pure elements in several intermetallic phases has been calculated by Sluiter [30], and for compatibility between assessments it is important to use the same values for the same structures and elements in different assessments. These have almost the same relevance for multi-component modeling as the lattice stabilities of the pure elements discussed in Section 1.1.

### 2.2. The Reciprocal Energy and Miscibility Gap

The Gibbs energy difference between the endmembers on the diagonals in a reciprocal system is very important:

$$\Delta G^{\text{recip}}_{\text{AB:CD}} = {}^{\circ}G_{\text{A:C}} + {}^{\circ}G_{\text{B:D}} - \left( {}^{\circ}G_{\text{A:D}} + {}^{\circ}G_{\text{B:C}} \right) \tag{8}$$

This reciprocal energy, $\Delta G^{\text{recip}}_{\text{AB:CD}}$, corresponds to the energy difference of the diagonals in the center of the reciprocal square, as shown in Figure 2 for two different cases of $\Delta G^{\text{recip}}_{\text{AB:CD}}$. In Figure 2a the diagonals have the same energy in the middle but in Figure 2b there is a difference, and if this is sufficiently large there is a miscibility gap, even if all excess parameters are zero.

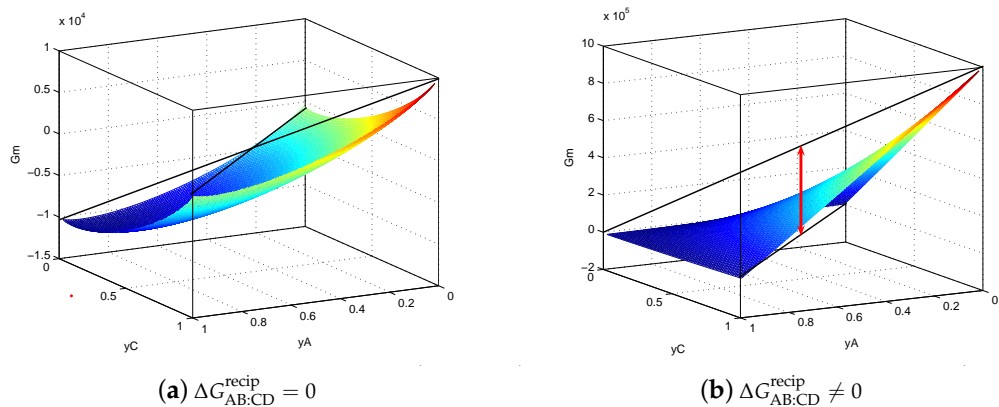

(**a**) $\Delta G^{\text{recip}}_{\text{AB:CD}} = 0$      (**b**) $\Delta G^{\text{recip}}_{\text{AB:CD}} \neq 0$

**Figure 2.** The Gibbs energy surface without (**a**) and with a reciprocal miscibility gap (**b**).

In many reciprocal systems the miscibility gap is real—for example, in the C-Fe-Ti system (see Section 4.1) there is one between austenite where C dissolves interstitially in FCC-Fe and the cubic TiC carbide, with considerable fraction of Va, both having the B1 structure. However, in other systems the miscibility gap can appear because a metastable endmember parameter is badly estimated. This can be problematic because it is not possible to suppress this miscibility gap using interaction parameters.

When modeling a single phase with the reciprocal model it is always possible to set three of the endmember energies to zero to plot a Gibbs energy surface, as in Figure 2b. If the Gibbs energies of the endmembers vary independently on $T$ there is a risk of forming a miscibility gap when $T$ varies. However, frequently one or more of the endmembers are metastable and can be estimated in a way

that prevents a reciprocal miscibility gap. For example if three are known, the fourth, $^{\circ}G_{\text{B:D}}$, can be estimated by forcing $\Delta G^{\text{recip}}_{\text{AB:CD}} = 0$:

$$\Delta G^{\text{recip}}_{\text{AB:CD}} = 0 \quad = \quad {}^{\circ}G_{\text{A:C}} + {}^{\circ}G_{\text{B:D}} - ({}^{\circ}G_{\text{A:D}} + {}^{\circ}G_{\text{B:C}}) \tag{9}$$

$$^{\circ}G_{\text{B:D}} \quad = \quad {}^{\circ}G_{\text{A:D}} + {}^{\circ}G_{\text{B:C}} - {}^{\circ}G_{\text{A:C}} \tag{10}$$

A reciprocal miscibility gap can usually be avoided in an assessment but may appear when combining two independent assessments of a phase modeled with sublattices to a ternary system because they share several endmembers. If experimentally there should be no miscibility gap, in this phase it can be quite difficult to suppress that without modifying the original assessments.

The reason to use the reciprocal model should be that A and B mix on one sublattice and C and D on another. We can compare this with a substitutional model mixing four species, AC, AD, BC and BD. In such a model there is no tendency to form a miscibility gap, but the crystallographic information that A and B mix on one sublattice and C and D on the other is lost.

### 2.3. The Reciprocal Model for Order–Disorder Transitions

A reciprocal model $(A,B)_a(A,B)_a$ can be used for a phase which can be totally disordered, i.e., with $y_{1,\text{A}} \equiv y_{2,\text{A}}$ in the disordered state. These endmember energies must be equal:

$$^{\circ}G_{\text{A:B}} = {}^{\circ}G_{\text{B:A}} \quad = \quad \frac{z}{2}u_{\text{AB}} \tag{11}$$

where $u_{\text{AB}}$ as the bond energy and $z$ is the number of bonds between the sublattices.

Ansara et al. [31] used a reciprocal model with different numbers of sites on the sublattices in their first assessments of the L1$_2$ ordering in FCC. Such a model means some atoms have nearest neighbors in the same sublattice and require additional interaction parameters related to the bond energy in that sublattice. This model for FCC ordering is now discouraged, as discussed in Section 4.5.2, and a four-sublattice model where all nearest neighbors are on different sublattices is recommended. As already discussed, such a four-sublattice model can be considered as a superposition of several reciprocal models.

The reciprocal model by itself can be used for an order–disorder transition when all nearest neighbor atoms are on the opposite sublattice—for example, an A2/B2 ordering modeled without vacancies. Figure 3 has several diagrams for a bond energy parameter $u_{\text{AB}} = -300R$. The diagrams show how the constituent fractions of the sublattices vary with $T$ or composition, the resulting configurational heat capacity and the 2nd order transition line in the phase diagram. In the model $^{\circ}G_{\text{A:B}} = {}^{\circ}G_{\text{B:A}} = \frac{z}{2}u_{\text{AB}}$ with $z = 8$. The maximum ordering $T_{o/d} = -\frac{z}{4}\frac{u_{\text{AB}}}{R} = 600$ K at the equi-atomic composition and in Figure 3d the configurational heat capacity at the equiatomic composition is plotted as a function of $T$. For $T > T_{o/d}$ it becomes zero as there is no SRO in this version of the reciprocal model; see Section 2.4. In Figure 3e the configurational heat capacity is plotted as a function of composition at $T = 400$ K in the ordered state and there is a peak in this heat capacity at the ideal ordering configuration which has been explained in [32,33].

### 2.4. An Approximation of the SRO Contribution

In an assessment of the Au–Cu system, Sundman et al. [3] derived a first order approximation of the SRO contribution to CEF by comparing the configurational entropy in a quasichemical model for BCC to that of a recipocal model. They could show that a reciprocal parameter, with a value related to the bond energy, gives a topologically correct phase diagram with separate maxima for ordered L1$_2$ and L1$_0$ also for a four-sublattice CEF model of Au–Cu.

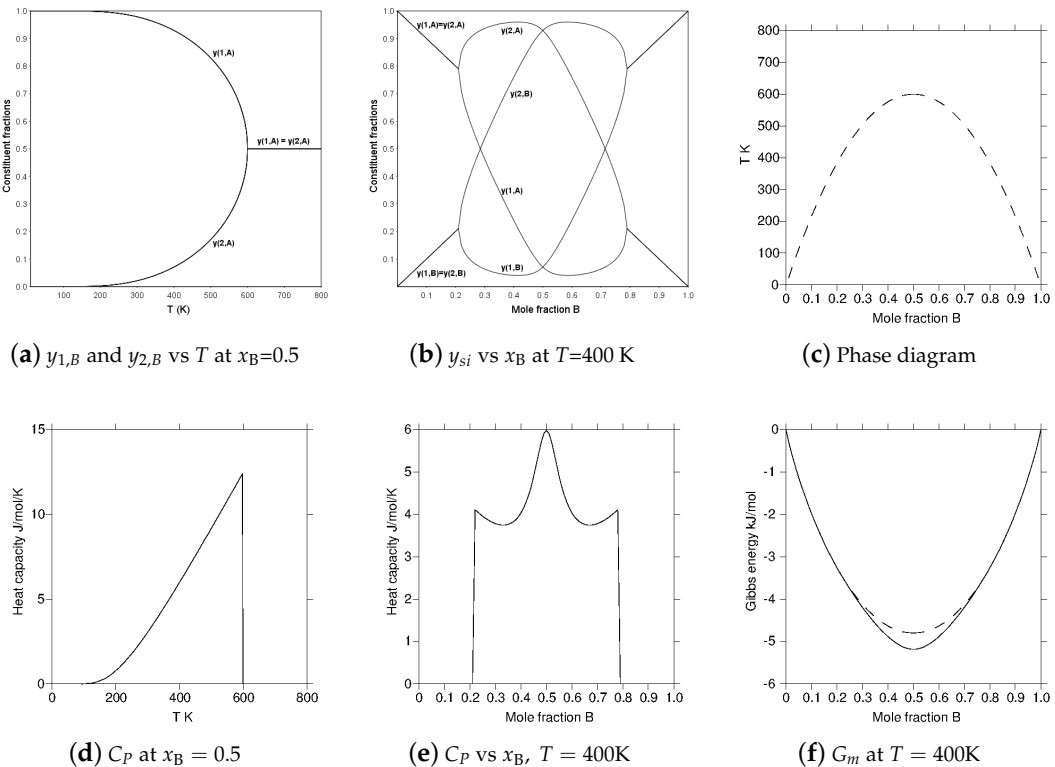

**(a)** $y_{1,B}$ and $y_{2,B}$ vs $T$ at $x_B$=0.5    **(b)** $y_{si}$ vs $x_B$ at $T$=400 K    **(c)** Phase diagram

**(d)** $C_P$ at $x_B = 0.5$    **(e)** $C_P$ vs $x_B$, $T = 400$K    **(f)** $G_m$ at $T = 400$K

**Figure 3.** Some calculated diagrams for order–disorder transformations for a reciprocal model with a single bond energy and no short range ordering (SRO). The 2nd order transition is dashed in (**c**); the dashed curve in (**f**) is without long range ordering (LRO).

The approximate SRO contribution for a reciprocal system (A,B)(A,B) was derived as:

$$\Delta G_M^{\text{SRO}} \quad \approx \quad -y_{1,A}y_{1,B}y_{2,A}y_{2,B}\frac{(\Delta G_{\text{AB:AB}}^{\text{recip}})^2}{zRT} \tag{12}$$

where $\Delta G_{\text{AB:AB}}^{\text{recip}}$ is the reciprocal Gibbs energy defined in Equation (8) when the constituents on the two sublattices are identical and $z$ is the number of nearest neighbors. $\Delta G_M^{\text{SRO}}$ depend on the constitution in the same way as the reciprocal parameter, $L_{\text{A,B:A,B}}$, and the approximate SRO contribution, $u_{\text{AB}}^{\text{SRO}}$, can be included in the reciprocal model parameter:

$$L_{\text{A,B:A,B}} \quad = \quad u_{\text{AB}}^{\text{SRO}} \approx -\frac{(\Delta G_{\text{AB:AB}}^{\text{recip}})^2}{zRT} \tag{13}$$

In a BCC phase $\Delta G_{\text{AB:AB}}^{\text{recip}} = \frac{z}{2}u_{\text{AB}}$ for a model with one atom/(unit cell) and at the temperature for the equiatomic order–disorder transition, $T_{o/d} \approx -\frac{z}{4}\frac{u_{\text{AB}}}{R}$, and these values can be inserted into Equation (13) to obtain the SRO parameter at the equiatomic order–disorder $T$:

$$u_{\text{AB}}^{\text{SRO}} \quad \approx \quad \frac{(\frac{z}{2}u_{\text{AB}})^2}{Rz\frac{z}{4}\frac{u_{\text{AB}}}{R}} = u_{\text{AB}} \tag{14}$$

Note that $u_{\text{AB}}$ must be negative to promote ordering and $u_{\text{AB}}^{\text{SRO}}$ is always negative in Equation (13). The bond energy $u_{\text{AB}}$ will normally depend on $T$ but the constant value evaluated at $T_{o/d}$ is a practical approximation at all $T$. There is a factor $\frac{1}{T}$ in Equation (13), which could give a reasonable decrease of the approximate SRO entropy at higher $T$, but such a factor would make the disordered state stable at low $T$. A constant $u_{\text{AB}}^{\text{SRO}}$ does not contribute to the configurational heat

capacity in the disordered state and will slightly overestimate the SRO at high T, but this can be compensated for by other model parameters.

The value of a reciprocal interaction parameter in an assessment will depend on many different kinds of experimental and theoretical data, and using a constant bond energy for SRO is a reasonable first approximation. In a CEF model, few parameters have a direct physical meanings but contain contributions from many different phenomena.

### 2.5. The Disordered Fraction Set

A phase $\alpha$ modeled with sublattices may have some properties which are independent of its constitution, i.e., $y_{s,i}^{\alpha}$, but others which may depend on the composition, i.e., the mole fractions $x_i^{\alpha}$, as discussed in [11]. The mole fractions can be calculated from the constituent fractions, $y$, using Equation (2). If the constituents are the same as the elements, and we have vacant sites, denoted Va, in some sublattices this becomes:

$$x_i^{\alpha} \quad = \quad \frac{\sum_s a_s^{\alpha} y_{s,i}^{\alpha}}{\sum_s a_s^{\alpha}(1 - y_{s,\mathrm{Va}}^{\alpha})} \tag{15}$$

Most implementations of the CEF model have introduced a "disordered fraction set," which grants a possibility of including a separate set of Gibbs energy parameters, $G_M^{\alpha/\mathrm{dis}}(x)$, which are added to the Gibbs energy for the ordered part, $G_M^{\alpha/\mathrm{ord}}(y)$ using the equation:

$$G_M^{\alpha}(y) \quad = \quad G_M^{\alpha/\mathrm{ord}}(y) + G_M^{\alpha/\mathrm{dis}}(x) - T\,{}^{\mathrm{cfg}}S_M^{\alpha/\mathrm{dis}}(x) \tag{16}$$

where the set of mole fractions, $x$, are calculated from the set of constituent fractions $y$, using Equation (15). The configurational entropy of the disordered fraction set, ${}^{\mathrm{cfg}}S_M^{\alpha/\mathrm{dis}}$, is subtracted because the configurational entropy is included in $G_M^{\alpha/\mathrm{ord}}$ using the constituent fractions. Further details can be found in [11].

The physical reason to add a disordered fraction set is that many phases with LRO have properties that are independent of the constitution and parameters, for these can be described using this set. However, the disordered fraction set also simplifies the construction of multi-component databases for phases with order–disorder transitions, such as BCC, FCC and HCP, which can exist both as completely disordered (with equal fractions on all sublattices) and as ordered. Parameters from binary and ternary assessments with only disordered BCC, FCC and HCP are simply added to the disordered fraction set when combined with systems assessed using the ordered model for these phases.

For historical reasons there is also a slightly different way to add the ordered and disordered parts:

$$G_M^{\alpha} \quad = \quad G_M^{\alpha/\mathrm{dis}}(x) + \Delta G_M^{\alpha/\mathrm{ord}}(y) \tag{17}$$

$$\Delta G_M^{\alpha/\mathrm{ord}} \quad = \quad G_M^{\alpha/\mathrm{ord}}(y) - G_M^{\alpha/\mathrm{ord}}(y = x) \tag{18}$$

With these equations, introduced by Ansara et al. [34], it is possible to assess the disordered part independently from the ordered. It can be useful when assessing a system with a phase such as FCC in Au–Cu that exists both as ordered and disordered. The parameters in $G_M^{\alpha/\mathrm{dis}}$ in Equation (17) are the complete description of the phase when it is disordered and the parameters in $G_M^{\alpha/\mathrm{ord}}$ can be assessed independently because they have no influence when the phase is disordered; i.e., $y = x$.

In addition to the sublattices for order–disorder, there may also be a sublattice for interstitial constituents in both the ordered and disordered fraction set.

### 2.6. CEF Calculations for Prototype FCC Ordering

In 1938, Shockley [35] presented a phase diagram (as in Figure 4a) for fcc ordering in an FCC system using a four-sublattice model with an ideal configurational entropy. This diagram has the wrong topology, as the maxima for the ordered phases, L1$_2$ and L1$_0$, are not separated and the reason

for this is that the SRO contribution is ignored. In 1951 Kikuchi [1] derived CVM, a mean field model for LRO and SRO and when computers were becoming available some 20 year later, several calculations of topologically correct phase diagram for FCC with separated maxima as in Figure 4b using CVM models were published [36–38]. For binary systems with simple lattices these calculations are trivial today, but for more complex structures, CVM models which may require a large number of clusters to describe SRO even, in systems with few components, there is still an interest in simpler models to describe SRO. An example of a simpler model is Figure 4c, a prototype FCC ordering with correct topology calculated using a four-sublattice CEF model with same bond energy as in (a) and (b) and with an SRO contribution according to Equation (14).

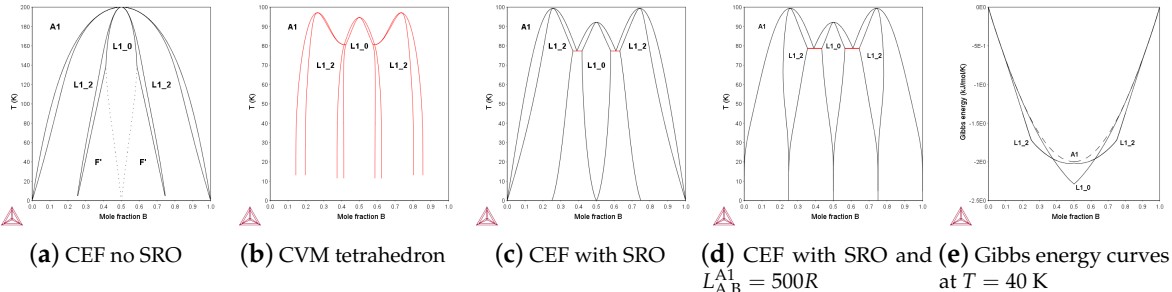

(**a**) CEF no SRO    (**b**) CVM tetrahedron    (**c**) CEF with SRO    (**d**) CEF with SRO and (**e**) Gibbs energy curves
                                                                    $L_{A,B}^{A1} = 500R$      at $T = 40$ K

**Figure 4.** These phase diagrams have been calculated with a single bond energy parameter $u_{AB} = -100R$. (**a**) The CEF model without SRO is used (as by Shockley), in (**b**) a tetrahedron CVM model and in (**c**) a CEF model with SRO according to Equation (14). Note also that the $T$ axis in (**a**) is double that in (**b**)–(**d**). In (**d**) and (**e**) a configuration independent parameter has been added.

The phase diagrams in Figure 4c,d use the four-sublattice CEF model with the parameters in Equation (19) and in Figure 4d,e also the interaction parameter in the disordered fraction set, $L_{A,B}^{A1}$, in Equation (20).

$$
\begin{aligned}
u_{AB} &= -100R \\
{}^{\circ}G_{A:A:A:B} = {}^{\circ}G_{A:A:B:A} = \cdots &= 3u_{AB} \\
{}^{\circ}G_{A:A:B:B} = {}^{\circ}G_{A:B:A:B} = \cdots &= 4u_{AB} \\
{}^{\circ}G_{A:B:B:B} = {}^{\circ}G_{B:A:B:B} = \cdots &= 3u_{AB} \\
L_{A,B:A,B:*:*} = L_{A,B:*:A,B:*} = \cdots &= u_{AB}^{SRO} = u_{AB} \\
L_{A,B}^{A1} &= 500R
\end{aligned}
\tag{19}
$$

$$
L_{A,B}^{A1} = 500R \tag{20}
$$

where $u_{AB}$ is the bond energy, $R$ the gas constant and the ${}^{\circ}G_{ijkl}$ are the energies for the endmember tetrahedra occupied with different sets of atoms. The $\cdots$ indicate all parameters with the same set of atoms permutated on the sublattices are equal. In a reciprocal parameter, $L_{A,B:A,B:*:*}$, the "*" indicates that the parameter is independent of the constituents in the sublattice. The configuration independent parameter in Equation (20) can be used to fit experimental data with no or minor effect on the topology.

The reciprocal parameters are an approximate contribution to the Gibbs energy due to SRO, and when set equal to the bond energy $u_{AB}$, in accordance with Equation (14) they give the correct topology in FCC as well, with separate L1$_2$ and L1$_0$ ordering regions. However, Equation (13) was derived by comparing the quasichemical model with a two-sublattice model for BCC and it means that in a four-sublattice CEF model for ordering in FCC the relation for SRO between the reciprocal parameter and reciprocal energy is:

$$
L_{A,B:A,B:*:*} = u_{AB}^{SRO} \approx -3\frac{(\Delta G_{AB:AB:*:*}^{recip})^2}{zRT_{o/d}} = u_{AB} \tag{21}
$$

when $\Delta G^{\text{recip}}_{AB:AB:*:*} = 2u_{AB}$, $z = 12$ and $T_{o/d} = -\frac{u_{AB}}{R}$. Several examples of ternary FCC ordered phase diagrams using this model are discussed in a paper by Kusoffsky et al. [39] and a simplified version is used in a commercial multi-component database for Ni-based superalloys [40].

### 3. Fe–Ni as an Example for Modeling with DFT

In the CE–CVM the SRO and LRO are described as Gibbs energy based on the atomic interactions. First the methodology of CE-CVM is explained and then the FCC phase in the Fe–Ni system is used as an example. This is then compared with the four-sublattice CEF model and with an assessment using also experimental data. The Fe–Ni system has a disordered FCC phase stable across the whole system at high $T$. At lower $T$ the Fe-rich side transforms to BCC, but on the Ni-rich side there is an order–disorder transition to an $L1_2$ ordered phase. This is also enhanced by a ferro-magnetic transition. Recent data [41,42] indicate also that there can be an $L1_0$ ordered phase at low $T$. This system is thus interesting to use as a comparison between CVM and CEF modeling using only DFT data and a full assessment using also experimental data.

### 3.1. Effective Cluster Interaction

The CVM requires the effective cluster interactions (ECIs), and these values are evaluated by using the cluster expansion method (CEM). In the CEM, each ordered structure, $R$ is represented as an arrangement of spin operators ($R = (\sigma_1, \sigma_2, \sigma_3, ..., \sigma_i, ...)$). Here, the spin operators ($\sigma_i$) correspond to each elemental species and index $i$ represents the atomic site position in the ordered structure. By using a combination of spin operators extracted from different sites, it is possible to represent clusters in the ordered structure. For example, $(\sigma_i, \sigma_j)$ is a pair cluster consisting of sites $i$ and $j$, and $(\sigma_i, \sigma_j, \sigma_k)$ is a three-body cluster consisting of sites $i$, $j$, $k$.

Furthermore, the correlation functions ($\xi_\alpha$) represented by Equation (22) are introduced by taking the average values of spin products for clusters such as points, pairs, three-bodies, and so on.

$$
\begin{aligned}
\xi_{point} &= \frac{1}{N_{point}} \sum_i \sigma_i \\
\xi_{pair} &= \frac{1}{N_{pair}} \sum_{i,j} \sigma_i \cdot \sigma_j \\
\xi_{tri} &= \frac{1}{N_{tri}} \sum_{i,j,k} \sigma_i \cdot \sigma_j \cdot \sigma_k
\end{aligned}
\tag{22}
$$

In the $\xi_\alpha$, $\alpha$ represents the type of cluster, and $N_{point}$, $N_{pair}$, $N_{tri}$ are the total numbers of points, pairs, and three-body sites in the structure, respectively. The energy of the regular structure can be expressed by the sum of the products of the correlation functions and ECIs ($J_{null}$, $J_{point}$, $J_{pair}$, $J_{tri}$, ...).

$$
E_R = J_{null} + J_{point}\xi_{point} + J_{pair}\xi_{pair} + J_{tri}\xi_{tri} + ...
\tag{23}
$$

Ideally, it is possible to reproduce the energy of any ordered structure strictly by using the infinite number of clusters in Equation (23); however, in the actual calculation, it is necessary to truncate it into a finite number of clusters. The cluster interactions tend to be stronger at shorter distances and weaker at longer distances. Therefore, a method to set a threshold for cluster size is generally adopted. The cluster of the maximum size is given as $\alpha_{max}$, and the product of the correlation function and ECIs are summed up for the sub-clusters contained in the $\alpha_{max}$.

$$
E_R = \sum_{\alpha}^{\alpha_{max}} J_\alpha \xi_\alpha
\tag{24}
$$

Since the correlation function is obtained from the ordered structure, and the total energy of the left side is obtained from DFT calculation, the unknown parameter is only ECIs. The ECIs can be determined by the method of least squares from the many number of relation expression of Equation (24).

### 3.2. CVM Calculation

Once ECIs are determined, the energy of any atomic configuration can be determined within the accuracy of the cluster expansion without DFT calculations. In addition, the free energy of mixing, including the entropy of the configuration, can be expressed as follows.

$$G = \sum_{\alpha}^{\alpha_{max}} J_\alpha \xi_\alpha - T \sum_{\alpha}^{\alpha_{max}} \gamma_\alpha S_\alpha \tag{25}$$

This equation is the second term of entropy added to Equation (24), and $\gamma_\alpha S_\alpha$ is the contribution of entropy from the cluster $\alpha$ using the Kikuchi–Barker coefficient [1,43]. The free energy is calculated by applying the variational method to Equation (25) so that it is minimized.

### 3.3. CEM and CVM Calculations for Fe–Ni

The DFT calculations were performed with the Vienna *ab initio* simulation package (VASP) [44,45] in which ion-electron correlation is described by projector augmented wave method (PAW) [46,47]. Correlation and exchange functions were given by the generalized gradient approximation proposed by Perdew et al. (PBE-GGA) [48]. We used pseudo-potentials for Fe and Ni with 8 and 10 of valence electrons, respectively. The k-point sampling meshes of the Brillouin zone for a FCC primitive cell were 1000 points and altered the *k*-points meshes so that the spanning of *k*-points (*dk*) are comparable. The Brillouin-zone integration was performed using the Methfessel–Paxton technique [49] and smearing of the electron was set to 0.1 eV. The plane-wave cutoff for the wave function was 350 eV. In all calculations, we assume an initial ferromagnetic order and the unit cells of the Fe–Ni ordered structures are fully relaxed with respect to the volume, shape of the unit cell and atomic positions. The formation enthalpy ($\Delta H$) of a structure is defined as the difference between its total energy and the concentration weighted total energy of pure Fe and Ni of FCC structure.

$$\Delta H(\mathrm{Fe}_{1-x}\mathrm{Ni}_x) = H(\mathrm{Fe}_{1-x}\mathrm{Ni}_x) - [(1-x)H(\mathrm{Fe}) + xH(\mathrm{Ni})] \tag{26}$$

For CEM and CVM calculation, we used the code developed in [50]. We used the ordinary tetrahedron-octahedron (TO) cluster as the basic cluster. An occupation variable of each possible site $i$ ($\sigma_i$) takes the value +1 if Fe occupies that site and 0 if Ni occupies that site.

Figure 5 shows the FCC structure and Table 1 lists the sub-clusters of the TO cluster. The specific values of the correlation functions of the L1$_0$ and L1$_2$ ordered phases are also shown in Table 1. The formation energy of each ordered phase is as follows from the values of these correlation functions and Equation (24).

$$
\begin{aligned}
E^{CEM}_{A1-\mathrm{Fe}} &= J_0 + J_1 + J_2 + J_3 + J_4 + J_5 + J_6 + J_7 + J_8 + J_9 + J_{10} \\
E^{CEM}_{L1_2\text{-Fe}_3\mathrm{Ni}} &= J_0 + \frac{3}{4}J_1 + \frac{1}{2}J_2 + \frac{3}{4}J_3 + \frac{1}{4}J_4 + \frac{1}{2}J_5 + \frac{1}{4}J_7 + \frac{1}{2}J_8 + \frac{1}{4}J_9 + \frac{1}{4}J_{10} \\
E^{CEM}_{L1_0\text{-FeNi}} &= J_0 + \frac{1}{2}J_1 + \frac{1}{6}J_2 + \frac{1}{2}J_3 + \frac{1}{6}J_5 + \frac{1}{6}J_8 \\
E^{CEM}_{L1_2\text{-FeNi}_3} &= J_0 + \frac{1}{4}J_1 + \frac{1}{4}J_3 \\
E^{CEM}_{A1-\mathrm{Ni}} &= J_0
\end{aligned}
\tag{27}
$$

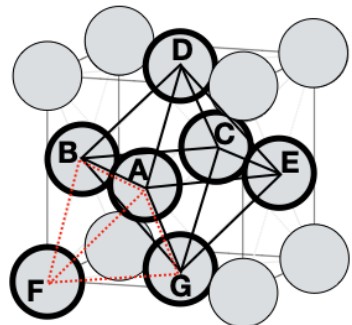

**Figure 5.** Tetrahedron and octahedron clusters on the FCC cell. Letters denote atomic sites and correspond to the third column of Table 1.

The optimal ECIs values for a set of 11 sub-clusters using 47 enthalpies of formation are the following (meV/atom):

$$
\begin{aligned}
J_0 &= -46.368 \\
J_1 &= -327.721 \\
J_2 &= 624.376 \\
J_3 &= -37.445 \\
J_4 &= -216.012 \\
J_5 \sim J_{10} &= 0
\end{aligned}
\tag{28}
$$

The ECIs with finite values consist of points, pairs and triangles. The ECIs of the other sub-clusters, including 4-6 points clusters, were given to 0.

The energies of $L1_0$, $L1_2$ from CEM were obtained from Equations (27) and (28). The directly calculated formation energies (eV/atom) of $L1_2$-Fe$_3$Ni, $L1_0$-FeNi and $L1_2$-FeNi$_3$ from DFT are as follows in units of eV/atom.

$$
\begin{aligned}
E^{DFT}_{L1_2\text{-Fe}_3\text{Ni}} &= -0.071679 \\
E^{DFT}_{L1_0\text{-FeNi}} &= -0.138536 \\
E^{DFT}_{L1_2\text{-FeNi}_3} &= -0.125748
\end{aligned}
\tag{29}
$$

The formation energies of $L1_2$-Fe$_3$Ni, $L1_0$-FeNi and $L1_2$-FeNi$_3$ from the ECIs are:

$$
\begin{aligned}
E^{CEM}_{L1_2\text{-Fe}_3\text{Ni}} &= -0.0620575 \\
E^{CEM}_{L1_0\text{-FeNi}} &= -0.124888 \\
E^{CEM}_{L1_2\text{-FeNi}_3} &= -0.137660
\end{aligned}
\tag{30}
$$

The predictive error [51], averaged over structures, becomes 17 meV/atom. The information of the crystal structures used in CEM is shown in Appendix A. The calculated phase diagram is shown in Figure 6a.

**Table 1.** Sub-clusters of the tetrahedron-octahedron (TO) clusters in FCC structure. Letters correspond to those in Figure 5. Specific values of correlation functions ($\xi_\alpha$) of L1$_2$ and L1$_0$ are also listed.

| $\alpha$ | Type | Site | $\xi_\alpha^{\text{L1}_2\text{-Fe}_3\text{Ni}}$ | $\xi_\alpha^{\text{L1}_0\text{-FeNi}}$ | $\xi_\alpha^{\text{L1}_2\text{-FeNi}_3}$ |
|---|---|---|---|---|---|
| 0 | null | – | 1 | 1 | 1 |
| 1 | single | A | 3/4 | 1/2 | 1/4 |
| 2 | pair | A, B | 1/2 | 1/6 | 0 |
| 3 | pair | A, C | 3/4 | 1/2 | 1/4 |
| 4 | 3 points | A, B, D | 1/4 | 0 | 0 |
| 5 | 3 points | A, B, C | 1/2 | 1/6 | 0 |
| 6 | 4 points | A, B, F, G | 0 | 0 | 0 |
| 7 | 4 points | A, B, C, D | 1/4 | 0 | 0 |
| 8 | 4 points | A, B, C, E | 1/2 | 1/6 | 0 |
| 9 | 5 points | A, B, C, D, E | 1/4 | 0 | 0 |
| 10 | 6 points | A, B, C, D, E, G | 1/4 | 0 | 0 |

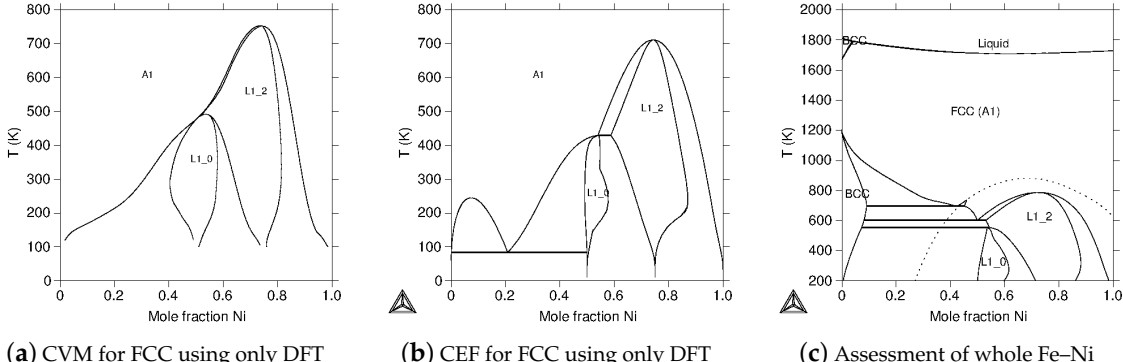

(**a**) CVM for FCC using only DFT     (**b**) CEF for FCC using only DFT     (**c**) Assessment of whole Fe–Ni

**Figure 6.** (**a**) The phase diagram calculated with CVM in the TO approximation using effective cluster interactions obtained from the CEM using DFT energy values in Equation (25) and in (**b**) the parameters in Equation (31). In (**c**) the whole Fe–Ni phase diagram calculated from an assessment by Ohnuma et al. [42].

### 3.4. CEF Calculation for Fe–Ni

Modeling based on CEF is described in detail in Lukas et al. [11]. For the FCC ordering with L1$_0$ and L1$_2$ on the Fe and Ni-rich sides we use a four-sublattice model with the following parameters taken from Equation (29):

$$
\begin{aligned}
{}^\circ G_{\text{Fe:Fe:Fe:Ni}} = \cdots &= -0.071679 * \text{ev2j} \\
{}^\circ G_{\text{Fe:Fe:Ni:Ni}} = \cdots &= -0.138536 * \text{ev2j} \\
{}^\circ G_{\text{Fe:Ni:Ni:Ni}} = \cdots &= -0.125748 * \text{ev2j} \\
L_{\text{Fe,Ni:Fe,Ni:*:*}} = \cdots &= -0.035 * \text{ev2j} \\
\text{ev2j} &= 96500
\end{aligned}
\tag{31}
$$

This set of parameters used just the first five values in Appendix A. From Equation (19) the reciprocal SRO parameter $L_{\text{Fe,Ni:Fe,Ni:*:*}}$ should be approximately $\frac{1}{4}$ of the endmember ${}^\circ G_{\text{Fe:Fe:Ni:Ni}}$ parameter. The ev2j value is the Faraday's constant to convert eV/atom to Joule/mol. A configuration independent contribution $L_{\text{Fe,Ni}} = 12,000$ J/mol was used to have the same maximum order–disorder $T$ for L1$_2$ as for the CVM calculation. Equation (16) was used to calculate the phase diagram in Figure 6b and the agreement with the CVM diagram is satisfactory. The small perturbation on the L1$_0$ phase is due to the formation of an F' phase, with the ordering (Fe:X:Ni:Ni) where X is close to Fe, formed by a second order transition from L1$_0$.

### 3.5. The Experimentally Assessed Fe–Ni System

The Fe–Ni system has been assessed with the Calphad method using the four-sublattice CEF model for the $L1_2$ and $L1_0$ ordering in the FCC phase by Cacciamani et al. [41], and more recently by Ohnuma et al. [42], who included new experimental data. In Figure 6c the latter assessment was used to calculate the phase diagram. The dashed line is the ferromagnetic transition line in FCC and there is a tri-critical point due to a magnetically induced miscibility gap in the FCC phase. The magnetic contribution to the Gibbs energy is based on an empirical model proposed by Inden [52] and explained in Lukas et al. [11], which depends on the experimental Curie temperature, $T_C$ and Bohr magneton number, $\beta$. With a CEF model it is possible to combine DFT data and experimental information to adjust endmember energies.

## 4. Examples Using DFT Data for CEF Models with LRO and SRO

A large number of the crystalline phases have some form of LRO, and some of them are explained in more detail below. In general it is simple to use DFT data in a CEF model when the crystalline structure is known. Phases with ionic constituents are described in Section 4.3, and how to model defects using CEF is described in Section 4.4.

For phases which always have LRO, the contribution to the Gibbs energy due to SRO is usually small and can be integrated in the LRO description. Thus SRO is only discussed in connection with phases which can also be stable while being totally disordered, such as FCC, BCC and HCP.

### 4.1. Interstitial Solutions, Carbides and Nitrides

A small atom like C may dissolve in the interstitial sites in metals, for example, the ferrite and austenite of Fe. This is a very important case as it gives unique properties to steel. The C atoms mix with vacancies in the octahedral interstitial sublattice in the austenite and this is a B1 structure. This means austenite is in fact the same phase as the cubic carbide in, for example, Ti-C. In the C-Fe-Ti system, austenite and the cubic TiC phase are thus modeled as the same phase. In the assessment by Ohtani et al. [53] it was described as a reciprocal system $(Fe,Ti)_1(Va,C)_1$ where the endmember (Fe:Va) is the pure iron and (Ti:C) the stable stoichiometric cubic carbide. The other two endmembers (Fe:C) are a metastable cubic carbide and (Ti:Va) a metastable FCC-Ti. In the C-Fe-Ti system there is a miscibility gap between Fe (with some dissolved C) and the TiC carbide (with a significant amount of Va).

Most carbides and nitrides can also be described using one sublattice for the metallic elements and one for C and N, sometimes also including Va and DFT calculations can be used to estimate the metastable endmembers. Sometimes more complex models are used, for example the $\kappa$ carbide, with a perovskite structure, was assessed in an in the Al-C-Fe system by Connetable et al. [54] using 4 sublattices for FCC on the Al-Fe side (which represent an $L1_2$ structure) and an interstitial sublattice site with C and Va. A full assessment would require 8 sublattices to describe also ordering for the C and Va. In a recent reassessment of this system by Zheng et al. [55] the same models for ordering were used with an improved fit to the experimental data.

### 4.2. Intermetallic Phases

There are a large number of intermetallic phases, Laves, A15, $\sigma, \mu, \chi$, etc., with different types of structures and solubility ranges. In some cases the structure is not even known but in most cases it is possible to define a CEF model which describes the stable composition range. But frequently the model has a larger composition range than the range of stability of the phase and there are often many endmembers that cannot be determined experimentally. Before DFT was available the number of crystallographic sites were often reduced to make it possible to estimate the endmember energies by extrapolations from the stable range. Even today it can be a cumbersome task to determine all endmembers in a multi-component system for a phase with more than three sublattices when there are

three or more components. As already mentioned, many of these endmembers may be mechanically unstable which means the value calculated by DFT is meaningless, see Section 5.2.1.

In order to combine separate assessments of systems including intermetallic phases in a database the endmember energies for the pure elements must be the same in all assessments in such a phase. As already mentioned several such energies for TCP phases have been published by Sluiter [30].

### 4.2.1. The $\sigma$ Phase and Other TCP Phases

The $\sigma$ phase has five sublattices but in many commercial databases the $\sigma$ phase is modeled using three sublattices, $(A,B)_{10}(B)_4(A,B)_{16}$ with a restricted composition range. Here the elements A are those with preferred FCC lattice, B those with preferred BCC lattice. The reason for this simplification was to reduce the number of endmembers and thus difficulties assessing endmembers values and to reduce calculation time.

Today this restriction is less valid, as the thermodynamic software are better and computer hardware is also much faster. Thus a five-sublattice model is the natural model to use to take into account the crystallographic structure for the $\sigma$ phase. In a multi-component system this may mean several 1000 endmembers but only a fraction of these are needed in the description of the stable range of the $\sigma$ phase because one can ignore those that have small chances to be a stable endmember as discussed in Section 5.2.

Many TCP phases such as $\mu, \chi$, A15, Laves, etc., are modeled differently in different assessments. This makes it difficult to combine the assessments to databases and the general recommendation is to use the crystallographically correct number of sublattices even if one can simplify the model in a particular system.

### 4.2.2. The Effective Bond Energy Formalism

An interesting new development is the Effective Bond Energy Formalism (EBEF) recently proposed by Dupin et al. [9]. EBEF assumes ideal mixing of the constituents on each sublattice but the endmember parameters are replaced by bond energy pairs between constituents in different sublattices. This can reduce the time for equilibrium calculations and the number of DFT calculations needed for endmember parameters multi-component systems. It also has the potential of improving the predictions from lower order to higher order systems but more work is needed to test this model.

### 4.3. CEF Models with Ionic Constituents

Oxygen behaves differently from C or N because it will normally capture electrons from the metallic elements. S is also prone to capture electrons and compounds with F, Cl etc are also strongly ionic but there are fewer solution phases with these elements. There has been several assessments for different oxide systems using CEF—for example, [56–63] using both experimental and DFT data.

In this context we will only look at some aspects of modeling the $UO_2$ phase with C1 structure which is important as nuclear fuel. The phase diagram for $UO_{2\pm x}$ shown in Figure 7a is calculated from an assessment by Guéneau et al. [64] as part of the development of the Thermodynamics of Advanced Fuels – International Database (TAF-ID) [65,66]. The model for the $UO_{2\pm x}$ phase with a C1 structure, in Figure 8a, takes into account the wide composition range and that U can have several valences and that a small amount of O can also occupy octahedral interstitial sites:

$$(U^{+3}, U^{+4}, U^{+5})_1 (O^{-2}, Va)_2 (Va, O^{-2})_1$$

where the third sublattice represent the interstitial sites. There is an enormous amount of experimental data on this system and there are also DFT calculations [60,67,68] of defects but so far there is no CVM based model which can consistently describe all data with the same accuracy as the CEF model. For simulations of the fuel behaviour it is also important that the model can describe the solubility of 10 or more fission products and equilibria with new phases formed during the burnup of the fuel.

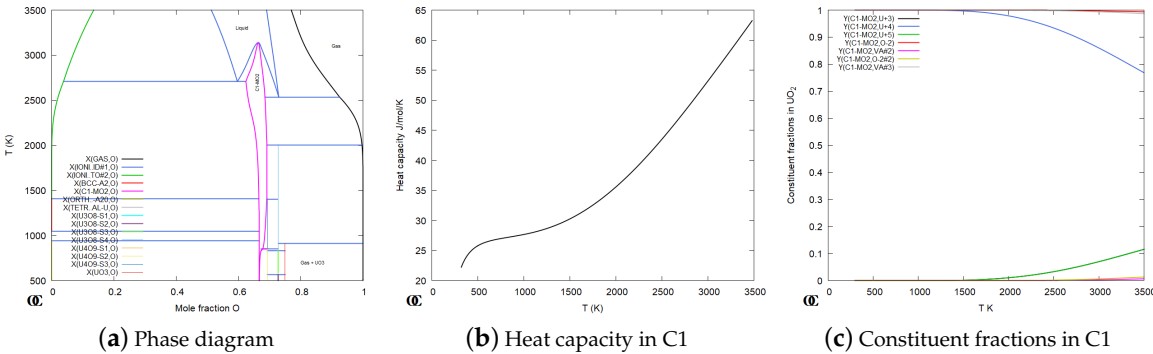

(**a**) Phase diagram　　　　(**b**) Heat capacity in C1　　　　(**c**) Constituent fractions in C1

**Figure 7.** (**a**) The phase diagram for the U-O system, (**b**) the heat capacity (with a significant contribution from the configurational entropy) and (**c**) the constituent fractions in C1 at ideal stoichiometry as a function of *T*.

The CEF model for $UO_{2\pm x}$ has 12 endmembers but only one of these is electrically neutral, all other has a net charge. To understand the relations between endmembers the prisms in Figure 8b,d can be helpful. They both represent the C1 phase but the endmembers are joined in different order. The endmembers are denoted "4OV," "4OO," "5VV," etc., to specify the constituents in all three sublattices. The shaded surfaces inside the prisms are the regions where the phase is electrically neutral. With some efforts one can figure out that the two separate shaded areas in Figure 8d are the same area as in Figure 8b.

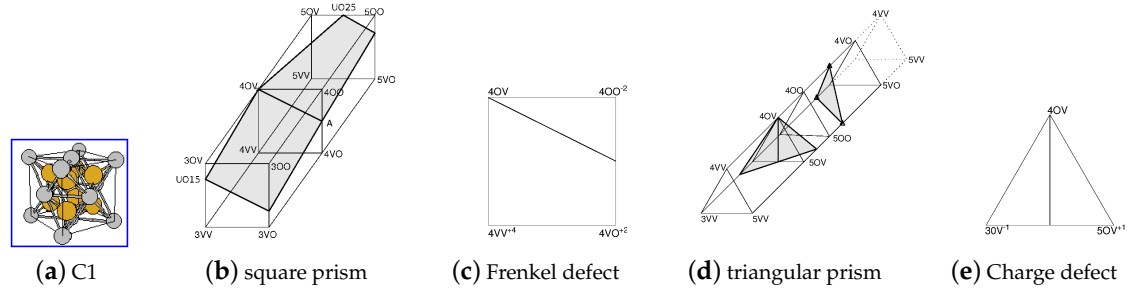

(**a**) C1　　　(**b**) square prism　　　(**c**) Frenkel defect　　　(**d**) triangular prism　　　(**e**) Charge defect

**Figure 8.** Graphical interpretation of the relations between endmembers and defects in $UO_2$ in Equations (32) and (33). The two prisms in (**b**) and (**d**) contain all 12 endmembers in slightly different arrangements. In (**c**) the endmembers related to the Frenkel defect are shown and in (**e**) are those related to the charge defect.

Figure 8 makes it possible to relate the Gibbs energies for different endmembers to properties that can be measured or calculated by DFT. For example Figure 8c illustrate a defect when one oxygen ion moves from the tetrahedral to the octahedral sublattice which can be related to the difference between endmember energies in Equation (32). The defect in Figure 8e represent a change of the valency of two U ions: $2\,U^{+4} = U^{+3} + U^{+5}$. The use of these defects is explained in more details in [69].

$$G^{\text{Frenkel}} = 0.5({}^{\circ}G_{U^{+4}:O^{-2}:O^{-2}} + {}^{\circ}G_{U^{+4}:Va:O^{-2}}) - {}^{\circ}G_{U^{+4}:O^{-2}:Va} \tag{32}$$

$$G^{\text{charge}} = 0.5({}^{\circ}G_{U^{+5}:O^{-2}:Va} + {}^{\circ}G_{U^{+3}:O^{-2}:Va}) - {}^{\circ}G_{U^{+4}:O^{-2}:Va} \tag{33}$$

### 4.4. CEF Models for Defects

There is no real difference between modeling defects and solubilities in a phase. Defects can be substitutional, interstitials or antisite and the latter two can be modeled as a reciprocal system. Traditionally very simple models like the Wagner-Schottky [70] have been used to handle small solubilities but it cannot be extended to multi-component systems, to higher fractions of defects or

when several defects may interact, as is the case for the defects in $UO_{2\pm x}$ in the previous section. The reciprocal model for the Laves phase was discussed in Section 2.1 and it can be extended to higher fractions of defects by additional parameters. Defect models and in particular thermal vacancies are discussed by Rogal et al. [71].

*4.5. Phases with Order–Disorder Transitions*

The FCC, BCC and HCP phases have a large number of ordered superstructures. Most of them, such as $D0_{22}$, $D0_{23}$, etc., can be treated as separate phases but in a few cases the equilibrium between the ordered and disordered structures are technically important and an accurate description requires that both the ordered and disordered phases are described with the same Gibbs energy model.

### 4.5.1. BCC Ordring Using Two Sublattices

The B2 phase is important in many alloys and the A2/B2 transition can be modeled with two sublattices. But if the Heusler phase, $L2_1$, is important one must have four-sublattices [8]. A particular problem with the B2 phase is that one frequently has vacancies on one side of the ideal ordering. For example in the Al-Ni system the B2 is stable in the middle as shown in Figure 9a and on the Al rich side, the Ni atoms are replaced by vacancies, not by Al atoms. This requires that one includes thermal vacancies already in the model for the disordered BCC phase as explained in the assessment of Al-Ni by Ansara et al. [34].

### 4.5.2. Never Model FCC Ordering with Two Sublattices

The two-sublattice model for FCC ordering using $(A,B)_{0.75}(A,B)_{0.25}$ is obsolete today. It cannot describe the $L1_0$ and in multi-component systems it requires a very complicated set of additional ternary and higher order parameters, related to the different binary bond energies. If these are not correct the "disordered state" with exactly the same constituent fractions on both sublattices will not exist. It is very easy to forget some of these parameters and end up with an FCC phase that is never completely disordered and this is not evident in a calculated phase diagram. A careful check must be made that the constituent fractions in the disordered state are identical in both sublattices.

### 4.5.3. The Four-Sublattice Model for Order–Disorder

The four-sublattice model

$$(A, B)_{0.25}(A, B)_{0.25}(A, B)_{0.25}(A, B)_{0.25}$$

can represent a tetrahedron in the FCC lattice, positions A-B-F-G in Figure 5. It is symmetrical with all nearest neighbors on another sublattice and it can describe the ordered $L1_2$ and $L1_0$ phases and calculate multi-component equilibria almost as fast as the unsymmetrical two-sublattice model if the software takes the symmetries into account. It is possible to add an interstital sublattice to this model representing the octahedral interstitial sites for C and N together with Va. New assessments of FCC ordering should always use the four-sublattice model to provide a better description of FCC ordering for future applications. The topology for the order–disorder transitions should be assessed also in the metastable composition ranges as for the FCC phase in Figures 6b and 9b in order to have reasonable extrapolations to higher order systems. If there is a need to be compatible with an old database using the two-sublattice model this can be assessed at the same time.

The ordered HCP phase can use the same four-sublattice model for B19 and $D0_{19}$ ordering. In addition to the Au–Cu system there are a few more assessments using the four-sublattice model, the Al-Ni system shown in Figure 9a from [72], Al-Ni-Pt by Lu et al. [73] and Fe–Ni by [41,42].

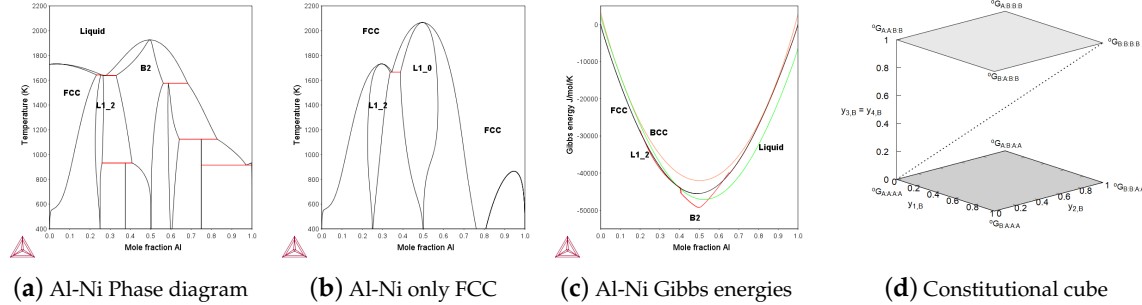

(**a**) Al-Ni Phase diagram (**b**) Al-Ni only FCC (**c**) Al-Ni Gibbs energies (**d**) Constitutional cube

**Figure 9.** (**a**) The Al-Ni phase diagrams with four-sublattice CEF ordering with SRO. (**b**) The phase diagram with just the FCC phase and (**c**) the Gibbs energy curves at 1500 K across the system. (**d**) An attempt to visuallize the 4D sublattice model for FCC in an A-B system. The lower left corner is pure A and upper right pure B and the dashed line represent the disordered state. Two constituent fractions, $y_{3,B} = y_{4,B}$, are equal on the vertical axis. The two shaded horizontal planes represent reciprocal systems for the other two sublattices. See the text for details.

The SRO contribution using four sublattices is obtained from the reciprocal energy, as described in Section 2.4, but as there are several such subsystems, the "constituent cube" in Figure 9d is an attempt to explain how this can be done. All the ordered structures $A_3B$, $A_2B_2$ and $AB_3$ are included in the Figure 9d and the dashed diagonal represents the disordered phase with all constituent fractions equal. The two shaded horizontal planes in Figure 9d represent reciprocal systems and the lower one has the endmembers (A:A:A:A), (A:B:A:A), (B:A:A:A) and (B:B:A:A) and the reciprocal Gibbs energy:

$$\Delta G^{\text{recip}}_{\text{X:X:A:A}} = {}^{\circ}G_{\text{B:B:A:A}} + {}^{\circ}G_{\text{A:A:A:A}} - \left({}^{\circ}G_{\text{A:B:A:A}} + {}^{\circ}G_{\text{B:A:A:A}}\right) = -2u_{\text{AB}} \tag{34}$$

using the endmember parameters in Equation (19). This reciprocal energy can be used in the reciprocal parameters to approximately describe SRO in this system and because all sites are equivalent there are two more unique reciprocal systems which give three reciprocal parameters to describe SRO in the four-sublattice CEF model: $L_{\text{X:X:A:A}}$, $L_{\text{X:X:A:B}} = L_{\text{X:X:B:A}}$ and $L_{\text{X:X:B:B}}$, where X indicate a sublattice with interaction. These can be different because the bond energy, $u_{\text{AB}}$, may be different for $x_B = 0.25, 0.5$ and 0.75. In Equations (19) and (31) $u_{\text{AB}}$ was assumed to be a constant.

For the BCC phase the four-sublattice model is needed for the D0$_3$ and Heusler phase, L2$_1$. It is more complicated than FCC because the BCC tetrahedron is unsymmetrical and the four-sublattice BCC model can also describe the B32 structure and there are four independent reciprocal parameters. The first assessment of the Al-Fe phase using four sublattices for BCC ordering and the ferro-magnetic ordering is shown in Figure 10a, assessed by Sundman et al. [23]. The calculated heat capacity including magnetism and configuration at $x_A = 0.3$ as a function of $T$ is shown in Figure 10b and in Figure 10c the variation of the constituent fractions. Several crystal structures and related endmembers based on the four-sublattice BCC model are explained in [8,74].

### 4.6. Modeling SRO in Liquids

Liquids are outside the scope of this paper, but the liquid phase is present in most phase diagrams and it is a kind of baseline for the crystalline phases. Metallic liquids with no or small SRO can usually be modeled as a substitutional regular solution, and there are three basic modeling techniques for SRO in liquids:

1.  The associated model which add constituents with a stoichiometry close to the SRO composition and include interactions between these and the constituents representing the elements, all mixing randomly. This overestimates the configurational entropy and requires many interaction parameters between the constituents. It includes substitutional regular solutions as a subset.

2.  The ionic two-sublattice liquid model [75] is a reciprocal model with cations on one sublattice and anions, vacancies and neutrals on the other. It is similar to a CEF model but the site ratios are variable. This model includes the substitutional regular solution and some associated models as subsets but can have problems with reciprocal miscibility gaps.

3.  The quasichemical model which is the simplest CVM based model has been modified by Pelton et al. [76] to exclude LRO and use only two bonds per constituent to avoid negative configurational entropy. The stoichiometries of the constituents are selected according to the SRO composition. It has some problems incorporating substitutional regular solutions.

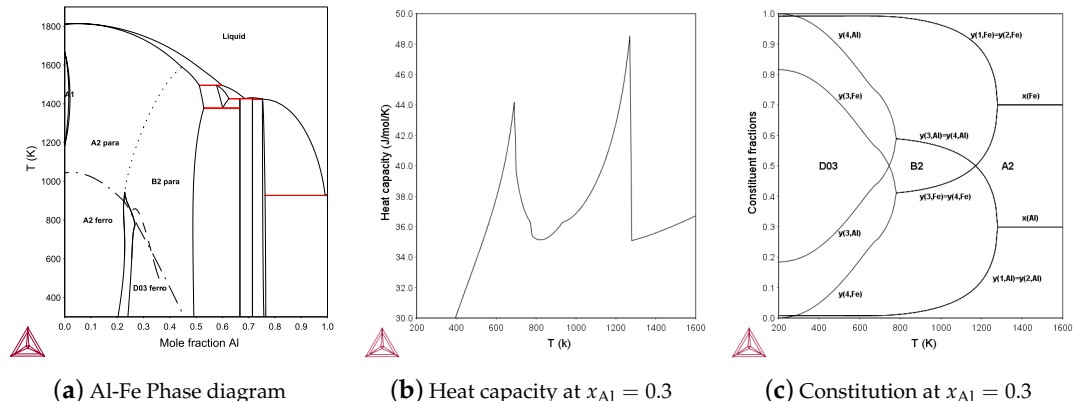

(**a**) Al-Fe Phase diagram      (**b**) Heat capacity at $x_{Al} = 0.3$      (**c**) Constitution at $x_{Al} = 0.3$

**Figure 10.** (**a**) The Al-Fe phase diagram with several order–disorder transformations in the BCC phase. (**b**) The heat capacity and in (**c**) the corresponding constituent fractions at $x_{Al} = 0.3$.

Each of these models has their advantages and disadvantages. More research based on both experiments and Molecular Dynamics is needed to find a liquid model which is really satisfactory.

## 5. Assessments Using CEF Models Including *ab initio* Data

In this section the methodology using CEF models is discussed using previous studies. There are several criteria for selecting a model for a crystalline phase when assessing a system:

1.  Physical soundness.
2.  Compatibility with existing assessments that should be combined with the new assessment.
3.  Amount of experimental and theoretical data.
4.  Extrapolations to higher order systems.

It is straightforward to combine results from DFT calculations with experimental information in CEF models. The DFT values are treated as any other experimental data on the thermodynamic properties such as enthalpies of formation or mixing, chemical potentials or heat capacities as well as phase diagram data when determining the model parameters. The most frequently used DFT results are energies of formation of endmembers at 0 K, but it is possible to use heat capacities and other properties calculated by DFT.

### 5.1. The Number of Sublattices to Use

There are still problems, including all crystallographic features in a sublattice model and in some cases simplifications being reasonable. For an assessment of a system which will not be used in any database, one should of course use the best physical model possible but if it is part of an ongoing database development there is little reason to use better models than already used in the database. However, new assessments of order–disorder transitions in FCC, BCC or HCP phases should always use four sublattices for the metallic sublattices, as discussed in Section 4.5.2. This will give improved extrapolations to multi-component systems, as shown in [8,73].

For TCP phases it is now possible to use the crystallographically correct number of sublattices combined with a disordered fraction set. As explained below, one can avoid calculating a large number of endmembers by DFT because only a few are needed to have a reasonable multi-component Gibbs energy surface for the stable phase.

In addition to the endmemer parameters, one can use interaction parameters in the disordered fraction set. Interaction parameters in the sublattices should be used only in phases with order–disorder transitions or CEF models describing phases with interstitials or defects, such as the Laves phase, where the two sublattices have very different coordination numbers, as discussed by Crivello et al. [77].

### 5.2. Reducing the Number of Endmember Parameters

A phase with several sublattices and many constituents will have several thousand endmembers in a multicomponent system. However, an endmember with a Gibbs energy far above the disordered Gibbs energy surface (sometimes called the convex hull), at the composition of the endmember, will never have any influence on this surface and can be ignored. Only those endmembers with a Gibbs energy below or close to the disordered Gibbs energy surface of the phases are needed.

In Figure 11a all 47 DFT calculations in Appendix A are plotted against the composition in mole fraction. For the CEF calculation in Figure 6b only those representing the lowest endmember energies are needed. In Figure 11b the energies of all 32 endmembers in a five-sublattice model for Cr–Fe are plotted from a calculation by Korzhavyi et al. [5], and in Figure 11c there are a large number of DFT calculations for several phases in the Re–Ta system from a paper by Palumbo [78]. Very few of them are needed for the lowest Gibbs energy curve for each phase using a CEF model and it should be possible to adapt the software for DFT calculations to use crystallographic information and already calculated bond energies to avoid spending effort on calculating properties of endmembers that are likely to be significantly above the "convex hull" of a phase.

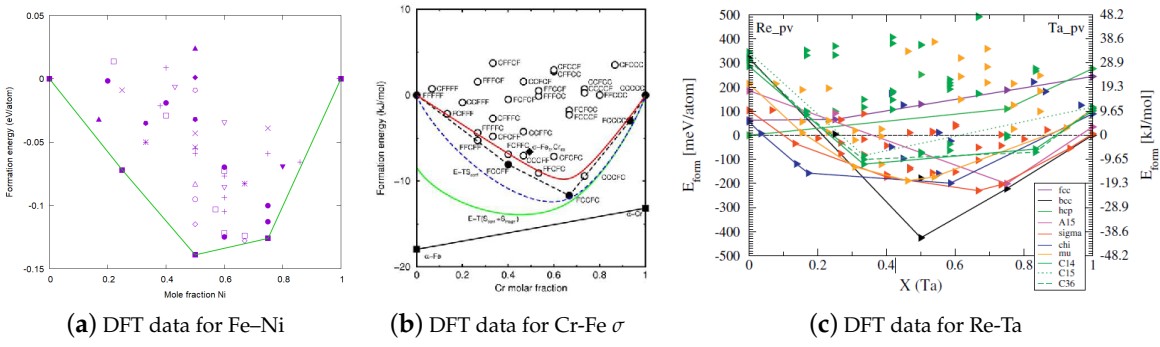

(**a**) DFT data for Fe–Ni          (**b**) DFT data for Cr-Fe $\sigma$          (**c**) DFT data for Re-Ta

**Figure 11.** (**a**) The 47 DFT calculations made in Fe–Ni. (**b**) The 32 DFT calculations made for the $\sigma$ phase in Cr-Fe by Korzhavyi et al. [5]. (**c**) Many DFT calculations for several phases in the Re-Ta system from Palumbo et al. [78].

### 5.2.1. Mechanically Unstable Endmembers

When calculating the formation energy for an endmember by DFT it is not unusual that the result represents a mechanically unstable structure. This is evident if the structure changes by relaxing it or if the calculated phonon frequencies of the endmember are negative. As an example, Figure 12 shows the calculation results for the Laves_C14 of $(Fe,Mo)_2(Fe,Mo)$. The calculation results of $Fe_2Mo$ and $Mo_2Fe$, which are the end members of $(Fe,Mo)_2(Fe,Mo)$, are shown in Figure 12a,b, respectively. $Fe_2Mo$, which is a stable structure, has positive frequencies for all phonon dispersions. On the contrary, due to the mechanically unstable, negative frequency, modes appear in the phonon dispersions for $Mo_2Fe$. Any thermodynamic value calculated for such a mechanically unstable endmember is meaningless. If the endmember is necessary for the model its values must be estimated in some other way, for example, using Equation (10), as discussed in Section 5.2.2.

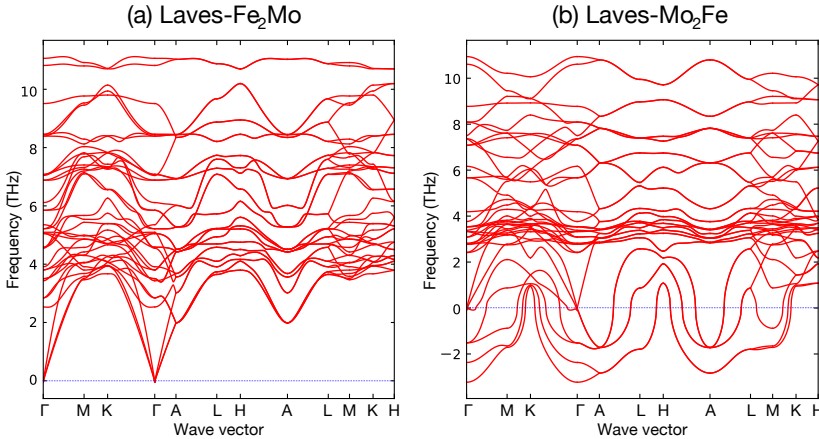

**Figure 12.** Phonon dispersions for Laves_C14 of (**a**) $Fe_2Mo$ and (**b**) $FeMo_2$.

There are cases when a phase is mechanically unstable at 0 K but can become stable at finite $T$, such as BCC for pure Ti. This is discussed in the paper by Antolin et al. [79] and also by [17].

### 5.2.2. Estimating Endmember parameters

Before DFT became an easily available tool, it was a complicated task to estimate metastable endmembers, in particular, endmembers such as $\,^\circ G_{\text{Fe:Fe}}^{\text{Laves\_C14}}$ which is shared among several different binaries and must have the same value in all. A long time ago it was agreed [80] to assume that the pure elements in the Laves phases should have an enthalpy difference between the stable pure element, GHSER, and the Laves phases equal to 5000 J/mol atoms, but this has now been replaced by the DFT calculations by Sluiter [30].

In a paper by Hillert et al. [81] it is explained how one can divide a complex CEF model into separate reciprocal systems and by setting $\Delta G_{\text{AB:CD}}^{\text{recip}} = 0$ and using Equation (10) to express the metastable or unstable endmembers as functions of three other endmembers. See also Section 4.3 for how to estimate some endmembers in the C1 phase.

### 6. Conclusions

Based on the explanations and discussions above we make the following conclusions:

- DFT is a powerful method to obtain many different kinds of information about materials from basic physical theories.
- The Calphad method uses a mean field theory to model the Gibbs energies of different phases, mainly based on experimental data, in order to calculate phase diagrams and extract the thermodynamic properties of a system.
- Results from DFT calculations fill an important gap in the data needed to determine thermodynamic properties for phase diagram calculations when experiments are very complicated or impossible.
- Increasing the number of DFT calculations and using complex CVM models does not generally give a better phase diagram when compared with experimental data than a simple CEF model based on much fewer DFT calculations.
- Data for stable and metastable phases for the elements and many binary compounds cannot be modified for each phase diagram. For the construction of a multicomponent database, such values must be agreed on at an early stage and can only be slightly modified even if new DFT or experimental data become available later.
- The success of the Calphad method is providing reasonable extrapolations to multi-component systems based on binary and ternary assessments, not giving exact values at a specific composition.

- The SRO contribution to the Gibbs energy of a phase can be approximated with CEF parameters for a phase with LRO.
- For a phase with order–disorder transition, the reciprocal model parameter gives a reasonable approximation to SRO, even when the phase has no LRO.
- Experimental and DFT data can be used together to determine the model parameters in a CEF model.
- The correct crystallography for the phases should be used in the Calphad models. It is possible to fit different sets of model parameters for a phase in order to be compatible with older databases as well.

**Author Contributions:** Conceptualization and methodology, M.E. and B.S.; software, M.H.F.S. and B.S.; writing B.S., M.E. and M.H.F.S.; editing and supervision M.S. and H.O. All authors have read and agreed to the published version of the manuscript.

**Funding:** H.O. gratefully acknowledges the financial support by JSPS KAKENHI (grant number, 16H02387) and M.S. acknowledges funding from the Competence Center Hero-m 2i.

**Acknowledgments:** Several discussions with Nathlie Dupin are gratefully acknowledged.

**Conflicts of Interest:** The authors declare no conflict of interest.

## Appendix A. DFT Results

**Table A1.** Arrangement of elements and total energies.

| Formula | Space Grope | Lattice Parameters | Sites | Total Energy (eV/Atom) |
|---|---|---|---|---|
| A1-Fe | $Fm\bar{3}m$ (225) | $a = b = c = 3.644$ | Fe (4a) 0.000 0.000 0.000 | $-8.0585024$ |
| $L1_2$-$Fe_3Ni$ | $Pm\bar{3}m$ (221) | $a = b = c = 3.585$ | Fe (3d) 0.500 0.000 0.000 Ni (1b) 0.500 0.500 0.500 | $-7.4830550$ |
| $L1_0$-FeNi | $P4/mmm$ (123) | $a = b = 2.521$ $c = 3.552$ | Fe (1a) 0.000 0.000 0.000 Ni (1d) 0.500 0.500 0.500 | $-6.9027660$ |
| $L1_2$-$FeNi_3$ | $Pm\bar{3}m$ (221) | $a = b = c = 3.544$ | Fe (1a) 0.000 0.000 0.000 Ni (3c) 0.500 0.000 0.500 | $-6.2428418$ |
| A1-Ni | $Fm\bar{3}m$ (225) | $a = b = c = 3.515$ | Ni (4a) 0.000 0.000 0.000 | $-5.4699578$ |
| FeNi | $R\bar{3}m$ (166) | $a = b = 2.464$ $c = 12.052$ $\gamma = 120.0$ | Fe (3b) 0.000 0.000 0.500 Ni (3a) 0.667 0.333 0.333 | $-6.8065125$ |
| $Fe_2Ni$ | $P\bar{3}m1$ (164) | $a = b = 2.533$ $c = 6.262$ $\gamma = 120.0$ | Fe (2d) 0.667 0.333 0.672 Ni (1a) 0.000 0.000 0.000 | $-7.2544473$ |
| $FeNi_2$ | $P\bar{3}m1$ (164) | $a = b = 2.513$ $c = 6.171$ $\gamma = 120.0$ | Fe (1a) 0.000 0.000 0.000 Ni (2d) 0.667 0.333 0.668 | $-6.4067287$ |
| $FeNi_2$ | $Immm$ (71) | $a = 2.517$ $b = 3.534$ $c = 7.540$ | Fe (2a) 0.000 0.000 0.500 Ni (4i) 0.500 0.500 0.334 | $-6.4476193$ |
| $Fe_3Ni$ | $R\bar{3}m$ (166) | $a = b = 2.480$ $c = 24.501$ $\gamma = 120.0$ | Fe (6c) 0.000 0.000 0.748 Fe (3a) 0.333 0.667 0.667 Ni (3b) 0.667 0.333 0.833 | $-7.4203427$ |
| FeNi | $R\bar{3}m$ (166) | $a = b = 2.524$ $c = 24.589$ $\gamma = 120.0$ | Fe (6c) 0.000 0.000 0.875 Ni (6c) 0.333 0.667 0.042 | $-6.8181210$ |

**Table A1.** *Cont.*

| Formula | Space Grope | Lattice Parameters | Sites | Total Energy (eV/Atom) |
|---|---|---|---|---|
| FeNi$_3$ | R$\bar{3}$m (166) | $a = b = 2.480$<br>$c = 24.148$<br>$\gamma = 120.0$ | Fe (3b) 0.667 0.333 0.833<br>Ni (6c) 0.333 0.667 0.916<br>Ni (3a) 0.000 0.000 0.000 | −6.1560090 |
| FeNi | C2/m (12) | $a = 8.174$<br>$b = 2.464$<br>$c = 4.262$<br>$\beta = 100.4$ | Fe (4i) 0.875 0.000 0.626<br>Ni (4i) 0.375 0.000 0.126 | −6.8229973 |
| FeNi$_3$ | C2/m (12) | $a = 8.187$<br>$b = 2.479$<br>$c = 4.275$<br>$\beta = 100.0$ | Fe (2c) 0.000 0.000 0.500<br>Ni (4i) 0.750 0.500 0.250<br>Ni (2b) 0.000 0.500 0.000 | −6.1755375 |
| FeNi | I4$_1$/amd (141) | $a = b = 3.566$<br>$c = 7.181$ | Fe (4b) 0.500 0.000 0.250<br>Ni (4a) 0.000 0.000 0.000 | −6.8589782 |
| FeNi$_3$ | I4/mmm (139) | $a = b = 3.499$<br>$c = 6.995$ | Fe (2b) 0.000 0.000 0.500<br>Ni (2a) 0.000 0.000 0.000<br>Ni (4d) 0.000 0.500 0.250 | −6.2167388 |
| FeNi | Pmmn (59) | $a = 2.536$<br>$b = 3.553$<br>$c = 5.071$ | Fe (2b) 0.500 0.000 0.627<br>Ni (2b) 0.500 0.000 0.126 | −6.8469643 |
| FeNi | P4/nmm (129) | $a = b = 2.447$<br>$c = 6.907$ | Fe (2c) 0.000 0.500 0.616<br>Ni (2c) 0.000 0.500 0.128 | −6.7401110 |
| Fe$_2$Ni$_3$ | R$\bar{3}$m (166) | $a = b = 2.515$<br>$c = 30.900$<br>$\gamma = 120.0$ | Fe (6c) 0.333 0.667 0.067<br>Ni (6c) 0.667 0.333 0.133<br>Ni (3a) 0.000 0.000 0.000 | −6.5908792 |
| Fe$_2$Ni$_3$ | R$\bar{3}$m (166) | $a = b = 2.488$<br>$c = 30.220$<br>$\gamma = 120.0$ | Fe (6c) 0.000 0.000 0.800<br>Ni (6c) 0.667 0.333 0.734<br>Ni (3a) 0.333 0.667 0.667 | −6.5399970 |
| Fe$_3$Ni$_2$ | C2/m (12) | $a = 8.140$<br>$b = 2.446$<br>$c = 6.040$<br>$\beta = 119.2$ | Fe (4i) 0.207 0.000 0.314<br>Fe (2c) 0.500 0.500 0.500<br>Ni (4i) 0.598 0.000 0.898 | −7.0140706 |
| Fe$_2$Ni$_3$ | C2/m (12) | $a = 8.324$<br>$b = 2.520$<br>$c = 6.171$<br>$\beta = 119.6$ | Fe (4i) 0.900 0.500 0.101<br>Ni (4i) 0.800 0.000 0.701<br>Ni (2c) 0.500 0.500 0.500 | −6.5823584 |
| Fe$_2$Ni$_3$ | C2/m (12) | $a = 8.319$<br>$b = 2.524$<br>$c = 6.162$<br>$\beta = 119.6$ | Fe (4i) 0.199 0.000 0.298<br>Ni (4i) 0.600 0.000 0.900<br>Ni (2c) 0.500 0.500 0.500 | −6.5991808 |
| Fe$_2$Ni$_3$ | C2/m (12) | $a = 7.929$<br>$b = 3.555$<br>$c = 5.616$<br>$\beta = 134.7$ | Fe (4i) 0.099 0.000 0.299<br>Ni (4i) 0.700 0.000 0.100<br>Ni (2d) 0.500 0.000 0.500 | −6.6103016 |
| FeNi$_4$ | I4/m (87) | $a = b = 5.534$<br>$c = 3.498$ | Fe (2b) 0.500 0.500 0.000<br>Ni (8h) 0.800 0.600 0.500 | −6.0571414 |
| Fe$_3$Ni$_2$ | Immm (71) | $a = 2.455$<br>$b = 3.467$<br>$c = 12.351$ | Fe (2c) 0.500 0.500 0.000<br>Fe (4i) 0.000 0.000 0.094<br>Ni (4i) 0.500 0.500 0.198 | −7.0520604 |

**Table A1.** *Cont.*

| Formula | Space Grope | Lattice Parameters | Sites | Total Energy (eV/Atom) |
|---|---|---|---|---|
| Fe$_2$Ni$_3$ | Immm (71) | $a = 2.503$<br>$b = 3.543$<br>$c = 12.637$ | Fe (4j) 0.000 0.500 0.601<br>Ni (4j) 0.500 0.000 0.700<br>Ni (2d) 0.500 0.000 0.500 | −6.6265112 |
| Fe$_2$Ni$_3$ | Immm (71) | $a = 2.508$<br>$b = 3.548$<br>$c = 12.637$ | Fe (4i) 0.500 0.500 0.699<br>Ni (4i) 0.000 0.000 0.600<br>Ni (2a) 0.000 0.000 0.000 | −6.5780444 |
| Fe$_4$Ni | I4/mmm (139) | $a = b = 2.454$<br>$c = 17.283$ | Fe (4e) 0.500 0.500 0.303<br>Fe (4e) 0.500 0.500 0.104<br>Ni (2a) 0.000 0.000 0.000 | −7.5419412 |
| Fe$_3$Ni$_2$ | I4/mmm (139) | $a = b = 2.461$<br>$c = 17.416$ | Fe (4e) 0.000 0.000 0.205<br>Fe (2a) 0.500 0.500 0.500<br>Ni (4e) 0.000 0.000 0.599 | −7.0416628 |
| Fe$_2$Ni$_3$ | I4/mmm (139) | $a = b = 2.513$<br>$c = 17.771$ | Fe (4e) 0.000 0.000 0.100<br>Ni (4e) 0.500 0.500 0.200<br>Ni (2b) 0.500 0.500 0.000 | −6.6305130 |
| Fe$_2$Ni$_3$ | I4/mmm (139) | $a = b = 2.514$<br>$c = 17.829$ | Fe (4e) 0.500 0.500 0.700<br>Ni (4e) 0.000 0.000 0.600<br>Ni (2a) 0.000 0.000 0.000 | −6.5752088 |
| FeNi | Cmmm (65) | $a = 3.559$<br>$b = 10.714$<br>$c = 3.561$ | Fe (2a) 0.500 0.000 0.000<br>Fe (4j) 0.000 0.666 0.500<br>Ni (4i) 0.500 0.667 0.000<br>Ni (2c) 0.000 0.000 0.500 | −6.8790688 |
| FeNi$_2$ | Cmmm (65) | $a = 3.547$<br>$b = 10.645$<br>$c = 3.548$ | Fe (4i) 0.500 0.666 0.000<br>Ni (4j) 0.000 0.667 0.500<br>Ni (2b) 0.500 0.000 0.000<br>Ni (2d) 0.000 0.000 0.500 | −6.4522342 |
| FeNi | C2/m (12) | $a = 4.275$<br>$b = 7.365$<br>$c = 4.271$<br>$\beta = 110.0$ | Fe (2a) 0.000 0.000 0.000<br>Fe (4h) 0.000 0.833 0.500<br>Ni (4g) 0.000 0.333 0.000<br>Ni (2d) 0.000 0.500 0.500 | −6.8185202 |
| FeNi$_4$ | C2/m (12) | $a = 4.361$<br>$b = 7.552$<br>$c = 4.345$<br>$\beta = 109.8$ | Fe (4h) 0.000 0.833 0.500<br>Ni (4g) 0.000 0.333 0.000<br>Ni (2d) 0.000 0.500 0.500<br>Ni (2a) 0.000 0.000 0.000 | −6.4018230 |
| FeNi | P$\bar{3}$m1 (164) | $a = b = 2.462$<br>$c = 11.976$<br>$\gamma = 120.0$ | Fe (2d) 0.333 0.667 0.167<br>Fe (1a) 0.000 0.000 0.000<br>Ni (2d) 0.333 0.667 0.667<br>Ni (1b) 0.000 0.000 0.500 | −6.7631747 |
| FeNi | Pmmm (47) | $a = 2.466$<br>$b = 3.462$<br>$c = 7.406$ | Fe (2r) 0.000 0.500 0.332<br>Fe (1d) 0.500 0.000 0.500<br>Ni (1e) 0.000 0.500 0.000<br>Ni (2s) 0.500 0.000 0.833 | −6.7725910 |
| Fe$_5$Ni | P4/mmm (123) | $a = b = 2.445$<br>$c = 10.406$ | Fe (2g) 0.000 0.000 0.667<br>Fe (2h) 0.500 0.500 0.826<br>Fe (1d) 0.500 0.500 0.500<br>Ni (1a) 0.000 0.000 0.000 | −7.6497458 |
| Fe$_2$Ni | P4/mmm (123) | $a = b = 2.468$<br>$c = 10.384$ | Fe (1a) 0.000 0.000 0.000<br>Fe (1d) 0.500 0.500 0.500<br>Fe (2g) 0.000 0.000 0.667<br>Ni (2h) 0.500 0.500 0.167 | −7.2382870 |

**Table A1.** *Cont.*

| Formula | Space Grope | Lattice Parameters | Sites | Total Energy (eV/Atom) |
|---|---|---|---|---|
| FeNi | P4/mmm (123) | $a = b = 2.454$ $c = 10.400$ | Fe (1b) 0.000 0.000 0.500 Fe (2h) 0.500 0.500 0.667 Ni (2g) 0.000 0.000 0.167 Ni (1c) 0.500 0.500 0.000 | −6.7959257 |
| FeNi$_6$ | R$\bar{3}$ (148) | $a = b = 6.569$ $c = 6.077$ $\gamma = 120.0$ | Fe (3a) 0.000 0.000 0.000 Ni (18f) 0.952 0.762 0.333 | −5.8977626 |
| Fe$_4$Ni$_3$ | R$\bar{3}$m (166) | $a = b = 2.458$ $c = 42.012$ $\gamma = 120.0$ | Fe (6c) 0.333 0.667 0.810 Fe (6c) 0.667 0.333 0.905 Ni (6c) 0.333 0.667 0.952 Ni (3a) 0.000 0.000 0.000 | −6.9524643 |
| Fe$_3$Ni$_4$ | Immm (71) | $a = 2.466$ $b = 3.485$ $c = 17.359$ | Fe (4i) 0.500 0.500 0.643 Fe (2a) 0.500 0.500 0.500 Ni (4i) 0.000 0.000 0.714 Ni (4i) 0.000 0.000 0.571 | −6.6858293 |
| Fe$_5$Ni$_3$ | P2/m (10) | $a = 6.012$ $b = 2.452$ $c = 6.040$ $\beta = 109.3$ | Fe (2n) 0.256 0.500 0.762 Fe (1b) 0.000 0.500 0.000 Fe (2m) 0.624 0.000 0.885 Ni (1h) 0.500 0.500 0.500 Ni (2m) 0.123 0.000 0.373 | −7.0955182 |
| FeNi$_3$ | I4/mmm (139) | $a = b = 3.497$ $c = 14.022$ | Fe (4e) 0.000 0.000 0.875 Ni (4c) 0.000 0.500 0.000 Ni (4e) 0.500 0.500 0.875 Ni (4d) 0.500 0.000 0.750 | −6.2303445 |
| Fe$_7$Ni$_2$ | Immm (71) | $a = 2.444$ $b = 7.358$ $c = 10.423$ | Fe (8l) 0.000 0.162 0.331 Fe (4j) 0.500 0.000 0.827 Fe (2d) 0.500 0.000 0.500 Ni (4g) 0.500 0.667 0.500 | −7.4754769 |

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
