# Peer review of "Calphad Modeling of LRO and SRO Using ab initio Data"

_metals, doi:10.3390/met10080998_

Round 1

Reviewer 1 Report

The paper discusses the integration of quantum mechanical calcualtion at DFT level into Calphad approaches to predict equilibrium properties of materials.

While I do not intend to contest the scientific validity of the work, I must stress that the presentation is rather confused and proceeds more by accumulation than by explanation. The paper oscillates from very technical sections to rather lengthy descriptions, with abrupt statements here and there. It is possible that this is due to some difficulty with the English language, and I would not want to blame the authors for it. But it certainly makes the reading difficult.

The conclusions are short and rather pointless, essentially stating that without experimental data DFT data alone are not enough for a Chalphad estimate.

I suggest an extensive editing of the text and a better organization of the contents.

Author Response

We are very grateful for the comments by the reviews, the paper covers a wide field and we believe that several scientists who use DFT calculations to develop thermodynamic databases for phase diagram calculations are not familiar with the models available. Thus we want to emphasize that one cannot adjust the properties of the pure elements or even binary compounds independently for each system because that means they cannot be combined and we explain the use of the CEF model for LRO and SRO modeling. We give references to papers where this is discussed in more detail. We have improved the connections between the different parts of the paper and for Figure 6 we have added some explanations. We have also expanded the conclusions based on the discussion in the paper.

Reviewer 2 Report

The paper entitled "Calphad Modeling of LRO and SRO using ab initio
data" and which has the objective of "An example how to use DFT data in CEF models will be given below together with some advice selecting the appropriate model for different phases" is somewhat confusing to this reviewer.

A large part of the paper is devoted to background explanations about unary databases and then about the CVM model. In contrast the CEF model, which is supposedly the topic of the paper is described in one short sentence "Modeling based on CEF is described in detail in Lukas et al.[6]."  So the authors should decide what is the topic of their study.

The main original result in this paper is presented in Fig. 6. Whereas some agreement is obtained between the CEF and CVM results, there remains considerable difference that is not discusssed at all. Furthermore, neither diagram agrees, even remotely, with the experimentally assessed diagram. Why is that?

There then follow a selection of previously published case studies which are reviewed in brief.  What is the purpose of these? The conclusions section offers no clue.

The authors must decide the purpose and the audience of this paper. Are they writing an introduction, in which case many details are missing and the advanced topics might be reduced. Or are they writing an advanced note for experts? In that case a lot of introductory detail should be removed and an advanced discussion added.  It is this reviewers belief that following this suggestion will greatly increase the potential impact of the paper.

Author Response

(The authors gave the same response as above.)

Round 2

Reviewer 1 Report

I do not think that a significant re-organization of the text has been really carried out, except for the conclusions. But if the authors think that it is enough to improve the readability of the paper, I will not object further.

Author Response

The manuscript is a compromise between many authors, the balance is delicate and we hope the article will be useful for some readers.  We are grateful for adding the conclusions.